# Sample Efficient Bayesian Learning of Causal Graphs from Interventions

**Zihan Zhou**[*]
Electrical and Computer Engineering
Purdue University
zhou1248@purdue.edu

**Muhammad Qasim Elahi**[*]
Electrical and Computer Engineering
Purdue University
elahi0@purdue.edu

**Murat Kocaoglu**
Electrical and Computer Engineering
Purdue University
mkocaoglu@purdue.edu

## Abstract

Causal discovery is a fundamental problem with applications spanning various areas in science and engineering. It is well understood that solely using observational data, one can only orient the causal graph up to its Markov equivalence class, necessitating interventional data to learn the complete causal graph. Most works in the literature design causal discovery policies with perfect interventions, i.e., they have access to infinite interventional samples. This study considers a Bayesian approach for learning causal graphs with limited interventional samples, mirroring real-world scenarios where such samples are usually costly to obtain. By leveraging the recent result of Wienöbst et al. [2023] on uniform DAG sampling in polynomial time, we can efficiently enumerate all the cut configurations and their corresponding interventional distributions of a target set, and further track their posteriors. Given any number of interventional samples, our proposed algorithm randomly intervenes on a set of target vertices that cut all the edges in the graph and returns a causal graph according to the posterior of each target set. When the number of interventional samples is large enough, we show theoretically that our proposed algorithm will return the true causal graph with high probability. We compare our algorithm against various baseline methods on simulated datasets, demonstrating its superior accuracy measured by the structural Hamming distance between the learned DAG and the ground truth. Additionally, we present a case study showing how this algorithm could be modified to answer more general causal questions without learning the whole graph. As an example, we illustrate that our method can be used to estimate the causal effect of a variable that cannot be intervened.

## 1 Introduction

Causal discovery refers to learning the underlying causal graph of a data generating process using a combination of observational and interventional data. This is a fundamental problem with applications across various areas including economics, genomics, meteorology, could computing, and etc. [Pearl, 2009, Hoover, 1990, King et al., 2004, Ikram et al., 2022, Runge et al., 2019]. In recent decades, the technological progress has facilitated the accumulation of vast quantities of observational data, i.e., data collected without perturbing the underling causal mechanisms. However, with only observational data, one can only recover the causal graph up to its Markov equivalence class (MEC) [Verma, 1991, Andersson et al., 1997]. In general, interventional data, i.e., data collected after a perturbation in

---

*Equal Contribution.

38th Conference on Neural Information Processing Systems (NeurIPS 2024).

the system, is needed to learn the whole graph. Therefore, many recent works aim to use both observational and interventional data [Choo et al., 2022, Squires et al., 2020, Shanmugam et al., 2015, He and Geng, 2008, Hauser and Bühlmann, 2014].

These works use perfect $do(X)$ intervention [Pearl, 2009] on a set of intervention targets $X$ and can be categorized along various dimensions. One such dimension is whether they are adaptive or non-adaptive. Non-adaptive approaches [Shanmugam et al., 2015], determine the intervention targets before any interventions (experiments), whereas adaptive algorithms [Squires et al., 2020, Choo et al., 2022] can suggest the next intervention targets based on previous experiments. Non-adaptive methods can parallelize experiments, while adaptive ones are sequential but demand fewer interventions to learn the entire graph [Choo and Shiragur, 2023]. These works provide theoretic guarantees that the experiments are sufficient to learn the graph. Jiang and Aragam [2024] established conditions under which latent causal graphs are nonparametrically identifiable and can be learned from unknown interventions.

Nevertheless, these studies operate under the assumption that an infinite amount of interventional data can be gathered after each intervention. However, in most real-world scenarios, collecting interventional data is considerably more challenging. For instance, while modern single-cell RNA sequencing technologies have facilitated the collection of vast amounts of gene expression data for downstream tasks [Stuart and Satija, 2019, Zhou et al., 2022], accurately perturbing each gene remains difficult [Chandrasekaran et al., 2024, Uhler and Shivashankar, 2017, Sharma et al., 2022]. Consequently, practical applications tend to favor interventional methods that require fewer data, although this aspect receives less attention. To bridge this disparity, our work assumes access to an infinite oracle of observational data, while only a finite number of interventional samples can be acquired for each intervention.

Among those that consider a limited number of interventional samples, the majority employ a Bayesian approach. This method offers several advantages, including the ability to make finer distinctions among different graphs and resilience to erroneous categorical decisions in the face of limited interventional samples. These studies can be categorized based on whether they assume a parametric structural causal model (SCM). Many works, such as those by Heckerman et al. [1997], Annadani et al. [2024], Toth et al. [2022], make certain assumptions about the SCM, such as additive Gaussian noise or linear causal models. However, their results will be inaccurate if the underlying SCM does not adhere to these assumptions. In contrast, the only non-parametric work [Greenewald et al., 2019] assumes that the ground truth causal graph is a forest of trees. Each undirected component of a tree with $n$ nodes contains $n$ Directed Acyclic Graphs (DAGs) in its MEC, which facilitates the tracking of the posterior of each possible DAG. For general undirected graphs, the MEC size could be exponential to $n$ [He et al., 2015, Meek, 1995] which makes the Bayesian posterior updating intractable. In this study, we adopt a non-parametric approach to mitigate the risk of erroneous outcomes resulting from assumptions about the SCM while mediating the problem of exponentially many DAGs in the MEC. We additionally assume causal sufficiency and faithfulness. Causal sufficiency asserts the absence of latent confounders, while faithfulness implies that every conditional independence relation that holds in the probabilistic model is entailed by the d-separation property of the causal graph. Given these assumptions, the PC algorithm [Spirtes et al., 2001] can be employed to identify all v-structures in the causal graph, followed by the application of Meek Rules [Meek, 1995] to derive the essential graph. The subgraph induced by the unoriented edges in the essential graph comprises multiple chordal chain components and can be oriented independently from other unoriented chain components and the oriented subgraph [Andersson et al., 1997]. Given our assumption of infinite observational data and that PC algorithm and Meek Rules are complete, i.e., they orient all the edges that can be identified from the data, our focus lies on fully orienting the undirected chordal chain graphs (UCCG) of the essential graph of the ground truth causal graph. The main contributions of our work are listed below:

- We study the causal discovery problem with limited interventional samples and propose an algorithm to solve this problem in an Bayesian approach. We analyze this algorithm and show in theory that it can learn the true causal graph with a high probability given a large enough number of interventional samples and calculate the convergence rate.

- We conduct experiments on simulated datasets with random chordal DAGs to compare our algorithm with other baseline methods. The results show that our algorithm outperforms other baselines in that it takes fewer interventional samples to receive the same accuracy.

Additionally, the performance of our algorithm is stable across different settings (order and density of the random DAGs).

- We discuss how we can modify our algorithm to answer more general causal questions without learning the whole graph using DAG sampling from the MEC. We show how we can estimate the posterior of a set to be the backdoor adjustment set of a causal query given the essential graph and some interventional data as an example. We further show through simulated experiments that we can modify our algorithm to estimate causal effects of variables that cannot be intervened.

**Outline of the paper:** In Section 2, we summarize the related works. In Section 3, we state the preliminary notations and background knowledge. In Section 4, we describe how we setup the initialization steps of the main algorithm. In Section 5, we describe the algorithm with details and analysis on the convergence to the ground truth. In Section 6, we discuss how our algorithm can be modified to answer more general causal questions with an example. In Section 7, we present the experiment results on simulated datasets and compare with baselines. In Section 8, we conclude with discussion of limitations and future extensions.

## 2 Related Works

We can roughly divide the causal discovery methods into 3 categories: non-adaptive, adaptive, and Bayesian. Non-adaptive methods design intervention targets for each experiment before retrieving interventional samples. Since no information is shared between experiments, they can be conducted in parallel. Eberhardt [2007] discussed non-adaptive methods as fixed searching strategies and they show that in the worst-case scenaris $\mathcal{O}(\log n)$ size $\frac{n}{2}$ interventions are necessary to orient the whole causal graph, regardless of the adaptivity of the algorithm. Hyttinen et al. [2013] show the connection between orienting causal graph via intervention and the concept of separating system from combinatorics. Ghassami et al. [2018] studied the problem of orienting the maximum expected number of edges with a budgeted number of size 1 interventions. Kocaoglu et al. [2017], Lindgren et al. [2018] studied the problem of non-adaptive causal graph learning while minimizing intervention costs when each vertex is assigned with a different intervention cost. Many of the early works in causal discovery are non-adaptive, and therefore they construct the theoretical basis of causal discovery algorithm. Separating system is also used in our proposed method.

Adaptive methods make the decision of next intervention based on the results of previous experiments. Less interventions are usually required than non-adaptive counterparts. Eberhardt [2007] compares adaptive strategies with non-adaptive ones and propose a heuristic algorithm. He and Geng [2008] proposed *MinMaxMEC* and *MaxEntropy* algorithms that select the next intervention target that minimizes the maximum $I$-MEC size or maximizes the intervention entropy respectively. Hauser and Bühlmann [2014] proposed *OptSingle* which calculates the target that minimizes the maximum number of unoriented edges in the interventional graph. Squires et al. [2020] show that any algorithm would take at least $\lceil \frac{\omega}{2} \rceil$ size 1 interventions to recover the whole graph and proposed a two-stage active learning algorithm. In the first stage, it finds the directed clique tree representation of the unoriented graph and orient the residuals separately in the second stage. Choo et al. [2022] show that a size 1 invervention set can orient the whole graph if and only if it cuts all the covered edges in the ground truth DAG, and propose a learning algorithm that intervenes on a $1/2-$separator in each experiment to iteratively decrease the size of the currently unoriented components.

The aforementioned works all assume the existence of an infinite intervention oracle. The Bayesian approach is usually used when this assumption is untenable. Heckerman et al. [1997] investigated learning Structural Causal Models (SCMs) under the assumption that each local likelihood is a combination of multinomial distributions and all variables are discrete. Similarly, Buntine [1994] discussed simple linear local likelihoods involving both discrete and continuous variables. Tong and Koller [2001] proposed an active learning algorithm for Bayesian network learning. Previous studies typically employed Markov Chain Monte Carlo (MCMC) for sampling Directed Acyclic Graphs (DAGs) to estimate posterior distributions over DAGs and function parameters, which grow exponentially with the size of the model. Acharya et al. [2018] studied the problem of distinguishing two causal models on the same graph with limited interventional samples. Nishikawa-Toomey et al. [2022] uses variational inference and GFlowNets [Bengio et al., 2023] to learn and parameterize the joint posterior distribution over DAGs and mechanisms while assuming linear model and equal

noise variances. Recent researches [Lorch et al., 2021, Charpentier et al., 2022, Cundy et al., 2021, Toth et al., 2022, Hägele et al., 2023] started leveraging gradient information for more efficient inference, while they suffer from poor inference quality. These studies all make specific parametric assumptions regarding functions or noises in the SCM, which lead to incorrect outcomes when these assumptions are violated. Kuipers et al. [2022] combines constraint-based methods and MCMC methods to efficiently search DAGs, but no guarantee of convergence is provided. To circumvent such issues, Greenewald et al. [2019] directly update the posterior of a specific subgraph containing the root when the graph is a tree without any constraint on the SCM. However in practice, causal graphs are usually more complicated than trees, which motivates our work.

## 3 Preliminaries

A causal graph $\mathcal{D} = (\mathbf{V}, \mathbf{E})$ is a directed acyclic graph (DAG) where the vertex set $\mathbf{V}$ represents a group of random variables. In the context of $\mathcal{D}$, a directed edge $(X, Y) \in \mathbf{E}$ from variable $X$ to variable $Y$ (denoted as $X \to Y$) signifies that $X$ serves as an immediate parent of $Y$. We denote the parent set of variable $Y$ by $\mathsf{pa}(Y)$. The cut at vertex set $\mathbf{X}$, denoted as $E[\mathbf{X}, \mathbf{V} \setminus \mathbf{X}]$, constitutes the set of edges between $\mathbf{X}$ and some vertex in the set $\mathbf{V} \setminus \mathbf{X}$. According to the Markov assumption, the joint distribution can be decomposed as $P(\mathbf{v}) = \prod_{i=1}^{n} P(v_i | \mathsf{pa}(X_i))$. A causal graph entails specific conditional independence (CI) relationships among variables via $d$-separation statements. The $d$-separation serves as a criterion to determine whether a set of variables $\mathbf{X}$ is independent of another set $\mathbf{Y}$, given a third set $\mathbf{Z}$. This approach links dependence to connectedness, meaning the existence of a connecting path, while it associates independence with the absence of connection or separation. A set of DAGs is deemed Markov equivalent when they entail identical conditional independence (CI) statements. All Markov equivalent DAGs must have the same adjacencies and unshielded colliders (or v-structures). An unshielded collider is a graph structure of the form $A \to B \leftarrow C$ where $A$ and $C$ are non-adjacent vertices. The set of all Markov equivalent DAGs is called a Markov Equivalence Class (MEC). We denote the set of all DAGs that is Markov equivalent to $\mathcal{D}$ as $[\mathcal{D}]$.

**Definition 1** (Faithfulness [Zhang and Spirtes, 2012]). *If the population distribution exhibits a conditional independence relation only when there exists a corresponding $d$-separation statement in the causal graph, then we say that the population distribution is faithful to the causal graph.*

The positivity assumption is fundamental for making causal inferences. It asserts that, in theory, every individual possesses a non-zero chance of being both exposed and unexposed [Hernan and Robins, 2020]. In many cases, we have access to abundant observational data, which allows for precise estimation of the true observational distribution. The no positivity violation assumption is merely needed for theoretical guarantees to hold while not required by the algorithm. We assume access to the observational distribution, and the observational distribution is faithful to the true causal graph, with no positivity violations.

A partially directed acyclic graph (PDAG) is a partially directed graph free from directed cycles. All Markov equivalent DAGs can be represented by a completed partially directed acyclic graph (CPDAG), denoted by $\mathcal{C}$. A DAG $\mathcal{D}$ can be represented by a CPDAG when they share the same set of adjacencies and unshielded colliders, and every oriented edge in the CPDAG is also present in $\mathcal{D}$ [Meek, 2013]. CPDAGs are chain graphs with chordal chain components [Andersson et al., 1997]. In graph theory, a chordal graph is one in which cycles of four or more vertices always contain an additional edge, called a chord. We denote the set of all chain components of a PDAG $\mathcal{P}$ as $CC(\mathcal{P})$.

An intervention on a subset of variables $\mathbf{W} \subseteq \mathbf{V}$, denoted by the do-operator $do(\mathbf{W} = \mathbf{w})$, involves setting each $W_j \in \mathbf{W}$ to $w_j$. For every intervention, we have an induced post-interventional graph denoted by $\mathcal{D}_{\overline{\mathbf{S}}}$ with incoming edges to vertices in $\mathbf{S}$ removed. We denote the interventional and observational distributions as $P_{\mathbf{w}}^{\mathcal{D}}(\mathbf{v})$ and $P^{\mathcal{D}}(\mathbf{v})$ respectively for a given DAG $\mathcal{D}$. Using the truncated factorization formula over the post-interventional graph ($\mathcal{D}_{\overline{\mathbf{S}}}$), we have the following:

$$P_{\mathbf{w}}(\mathbf{V}) := P(\mathbf{V} \mid do(\mathbf{W} = \mathbf{w})) = \prod_{\mathbf{v} \notin \mathbf{W}} P(v | \mathsf{pa}(v)) \tag{1}$$

Where $\mathbf{v}$ must be consistent with the intervention $do(\mathbf{W} = \mathbf{w})$. Consider a set of intervention targets $\mathcal{S} = \{\mathbf{S}_1, \mathbf{S}_2, ..., \mathbf{S}_n\}$ where each $\mathbf{S}_i \subseteq \mathbf{V}$ for all $i \in [n]$. For every target $\mathbf{S}_i \in \mathcal{S}$, we define the collection of all possible cut configurations, i.e., all possible orientations of edges in $E[\mathbf{S}_i, \mathbf{V} \setminus \mathbf{S}_i]$, as $\mathsf{C}_k(\mathbf{S}_i)$ for all $k \in [n_{\mathbf{S}_i}]$. Also note that $n_{\mathbf{S}_i} \leq 2^{|\mathbf{S}_i| d_m}$ where $d_m$ is the maximum degree of the graph.

Additionally, for bounded size intervention targets i.e., $|\mathbf{S}_i| \leq k$, we have $n_{\mathbf{S}_i} \leq 2^{kd_m}$. We use the notation $P_{\mathbf{s}_i}^{\mathsf{C}_k(\mathbf{S}_i)}(\mathbf{V})$ to represent the interventional distribution in $\mathcal{E}(\mathcal{D})$ with the cut configuration $C_k(\mathbf{S}_i)$. Also we use the notation $\mathsf{C}^*(\mathbf{S}_i)$ for the cut configuration in the true DAG $\mathcal{D}^*$. Given an intervention set $\mathbf{S}_i$, if we assume perfect intervention, one can recognize all the edges adjacent to vertices in $\mathbf{S_i}$ and further apply Meek Rules. The resulting PDAG is called an interventional essential (I-essential) graph of $\mathcal{D}$ denoted as $\mathcal{E}_{\mathbf{S_i}}(\mathcal{D})$. When $\mathbf{S_i}$ is empty, it is the observational essential graph. I-essential graphs are also known to be maximally oriented partially directed acyclic graphs (MPDAGs). We denote an MPDAG as $\mathcal{M}$ and the I-essential graph with $C_k(\mathbf{S}_i)$ as $\mathcal{M}_{C_k(\mathbf{S}_i)}$. MPDAGs are also chain graphs with chordal components [Hauser and Bühlmann, 2012]. We denote the KL divergence between two distributions $p, q$ as $D_{KL}(p||q)$.

# 4  Algorithm Initializations

Assume the true DAG is $\mathcal{D}^*$, our algorithm aims to return a most 'probable' causal graph $\mathcal{D}$ given $N$ interventional samples. With the access to infinite observational data, we can retrieve the essential graph $G = \mathcal{E}(\mathcal{D}^*)$ by PC algorithm [Spirtes et al., 2001] and joint distribution $P(\mathbf{V})$. Here we describe how we compute the separating system of $G$ and the causal effect of a given intervention set.

## 4.1  Separating System

As mentioned in Section 2, separating system plays the key role for non-adaptive intervention design of causal discovery. Roughly speaking, a separating system on a set of elements is a collection of subsets such that for every pair of elements from the set, there exists at least one subset which contains exactly one element from the pair. Consider an undirected complete graph $G$ of $n$ vertices indexed as $1, ..., n$, a separating system $\mathcal{S}$ on $[n]$ would cut every edge of $G$. In the worst case, a separating system is needed to learn the causal graph with $G$ as its essential graph [Shanmugam et al., 2015]. If we further bound the size of each $\mathbf{S} \in \mathcal{S}$ such that $|\mathbf{S}| \leq k, k < \frac{n}{2}$, the resulting set is called an $(n, k)$-separating system. We provide the formal definition below:

**Definition 2** ($(n, k)$-Separating System [Katona, 1966, Wegener, 1979]). *An $(n, k)$-separating system on $[n]$ is a set of subsets $\mathcal{S} = \{\mathbf{S}_1, \mathbf{S}_2, ..., \mathbf{S}_m\}$ such that $|\mathbf{S}_i| \leq k$ and for every pair $i, j$ there is a subset $\mathbf{S} \in \mathcal{S}$ such that either $i \in \mathbf{S}, j \notin \mathbf{S}$ or $i \notin \mathbf{S}, j \in \mathbf{S}$.*

We discuss the detailed steps of how we construct the $(n, k)$-separating system in Appendix D.

## 4.2  Enumerating Causal Effects

To use a Bayesian approach, we need to construct a set of disjoint events and track their posteriors. Here we consider the events as interventional causal effects when we intervene on a set of vertices. By the assumption of faithfulness, Lemma 1 shows that the post-interventional distribution is determined by the edge configurations that are adjacent to vertices in the intervention set. Perkovic [2020] proposed a formula is to identify any causal effect in an MPDAG. One can enumerate all valid edge cuts and use the identification formula to calculate the post-interventional distributions. Here we propose a simple Algorithm 3 to enumerate through all possible configurations of a given set $\mathbf{S}$ and calculate the post-interventional distribution $P_{\mathbf{s}}^{C(\mathbf{S})}(\mathbf{V})$ via DAG sampling. For each candidate configuration $C_k(\mathbf{S})$, we check if it contains invalid structures, i.e. unshielded colliders or cycles. If it is valid, we apply Meek Rules to get the MPDAG $\mathcal{M}_{C_k(\mathbf{S})}$ which is an I-essential graph $\mathcal{E}_{\mathbf{S}}(\mathcal{D}^*)$ for some DAG $\mathcal{D}_{C_k(\mathbf{S})} \in [\mathcal{D}^*]$ that is consistent with $C_k(\mathbf{S})$. The chain components of $\mathcal{M}$ are chordal graphs and can be oriented independently [Hauser and Bühlmann, 2012]. To efficiently calculate the interventional distribution, we use a DAG sampler [Wienöbst et al., 2023] to sample a DAG for each chain component and replace the edges in $\mathcal{M}_{C_k(\mathbf{S})}$ with the arcs in the sampled DAG to get a fully oriented graph $\mathcal{D}$. The sampling process takes linear time. Given the DAG $\mathcal{D}$, the interventional distribution could be calculated by using the Equation 1 over the DAG.

# 5  Algorithm Design and Analysis

In the previous section, we discuss two key steps for the algorithm initialization, which include the construction of $(n, k)$-separating systems and enumerating causal effects for all possible cutting edge

configurations for all the intervention targets in the separating system. For a given intervention target $\mathbf{S}_i$, we utilize the following result that under the faithfulness assumption, shows that we have a unique interventional distribution for every cutting edge orientation $\mathsf{C}_j(\mathbf{S}_i)$.

**Lemma 1.** *[Elahi et al., 2024] Assume that the faithfulness assumption holds and $\mathcal{D}^*$ is the true DAG. For any DAG $\mathcal{D}_1 \neq \mathcal{D}^*$, if $P_{\mathbf{s}}^{\mathcal{D}_1} = P_{\mathbf{s}}^{\mathcal{D}^*}$ for some $\mathbf{S} \subseteq \mathbf{V}$, they must share the same cutting edge orientation $\mathsf{C}(\mathbf{S})$.*

Lemma 1 allows us to uniquely orient cutting edges for every intervention target in the separating system, which in turn orients every edge and gives us the true DAG. We use the notation $D_{ab}^{\mathbf{s}_i} := D_{KL}(\,P_{\mathbf{s}_i}^{\mathsf{C}_a(\mathbf{S}_i)}(\mathbf{V}) || P_{\mathbf{s}_i}^{\mathsf{C}_b(\mathbf{S}_i)}(\mathbf{V})\,) > 0 \,\forall a \neq b\,,\,\forall a, b \in [n_{\mathbf{S}_i}]$ and $D^{\mathbf{s}_i} := \min_{\forall a \neq b\,,\,\forall a, b \in [n_{\mathbf{S}_i}]} D_{ab}^{\mathbf{s}_i}$. The idea is to use the interventional data $do(\mathbf{S}_i = \mathbf{s}_i)$ to determine the true cutting edge configuration $\mathsf{C}^*(\mathbf{S}_i)$ and repeat this for all intervention targets in the separating system. Suppose we have access to $m_{\mathbf{s}_i}$ i.i.d. samples from the intervention $do(\mathbf{S}_i = \mathbf{s}_i)$, which we denote as $\mathsf{Data}_{do(\mathbf{s}_i)} = \{\mathbf{v}_1, \mathbf{v}_2, ..., \mathbf{v}_{m_{\mathbf{s}_i}}\}$. We define the posterior probabilities of $\mathsf{C}_j(\mathbf{S}_i)$ for all possible cutting configurations at $\mathbf{S}_i$, i.e., for all $j \in [n_{\mathbf{S}_i}]$ where $n_{\mathbf{S}_i} \leq 2^{kdm}$ as follows:

**Definition 3.** *(Posterior Probabilities of Cutting Edge Configurations) Consider an intervention target $\mathbf{S}_i$ and a collection of all possible cutting edge configurations $\mathsf{C}_j(\mathbf{S}_i)$ for all possible cutting configurations at $\mathbf{S}_i$, i.e., for all $j \in [n_{\mathbf{S}_i}]$ and the interventional dataset of i.i.d. samples $\mathsf{Data}_{do(\mathbf{s}_i)} = \{\mathbf{v}_1, \mathbf{v}_2, ..., \mathbf{v}_{m_{\mathbf{s}_i}}\}$. We define probabilities of Cutting Edge Configurations as:*

$$P(\mathsf{C}_j(\mathbf{S}_i) \mid \mathsf{Data}_{do(\mathbf{s}_i)}) = \frac{P_{\mathbf{s}_i}^{\mathsf{C}_j(\mathbf{S}_i)}(\mathbf{v_1}, \mathbf{v_2}, ..., \mathbf{v}_{m_{\mathbf{s}_i}})\,p_j}{\sum_{a=1}^{n_{\mathbf{S}_i}} P_{\mathbf{s}_i}^{\mathsf{C}_a(\mathbf{S}_i)}(\mathbf{v_1}, \mathbf{v_2}, ..., \mathbf{v}_{m_{\mathbf{s}_i}})\,p_a} \quad \forall j \in [n_{\mathbf{S}_i}]\,,\,\forall \mathbf{S}_i \in \mathcal{S} \quad (2)$$

*where $p_a$ for all $a \in [n_{\mathbf{S}_i}]$ are the set of priors such that $\sum_{a=1}^{n_{\mathbf{S}_i}} p_a = 1$.*

Given the fact that the interventional samples are i.i.d we can rewrite the posterior probabilities as

$$P(\mathsf{C}_j(\mathbf{S}_i) \mid \mathsf{Data}_{do(\mathbf{s}_i)}) = \frac{\prod_{k=1}^{m_{\mathbf{s}_i}} P_{\mathbf{s}_i}^{\mathsf{C}_j(\mathbf{S}_i)}(\mathbf{v}_k)\,p_j}{\sum_{a=1}^{n_{\mathbf{S}_i}} \prod_{k=1}^{m_{\mathbf{s}_i}} P_{\mathbf{s}_i}^{\mathsf{C}_a(\mathbf{S}_i)}(\mathbf{v}_k)\,p_a} \quad \forall j \in [n_{\mathbf{S}_i}]\,,\,\forall \mathbf{S}_i \in \mathcal{S} \quad (3)$$

**Assumption 1.** *We assume access to the observational distribution that is faithful to the true causal graph. Furthermore, we have $\left|\log \frac{P_{\mathbf{s}_i}^{\mathsf{C}_a(\mathbf{S}_i)}(\mathbf{v})}{P_{\mathbf{s}_i}^{\mathsf{C}_b(\mathbf{S}_i)}(\mathbf{v})}\right| \leq \beta$ for every intervention set $\mathbf{S}_i \in \mathcal{S}$ and for all $a, b \in [n_{\mathbf{S}_i}]$.*

The second half of the assumption can be seen as a version of the positivity assumption commonly used in causal discovery and many other applications. We show that for the problem of learning the true cutting edge configuration, the posterior is consistent, i.e., as the number of samples $m_{\mathbf{s}_i} \to \infty$, the posterior of the true cutting edge configuration given the data converges to 1 with high probability.

**Lemma 2.** *(Posterior Consistency) Consider an intervention target $\mathbf{S}_i \in \mathcal{S}$ and the corresponding true cutting-edge configuration $\mathsf{C}^*(\mathbf{S}_i)$. As the number of samples $m_{\mathbf{s}_i} \to \infty$ in $\mathsf{Data}_{do(\mathbf{s}_i)} = \{\mathbf{v}_1, \mathbf{v}_2, \ldots, \mathbf{v}_{m_{\mathbf{s}_i}}\}$, the posterior of the true cutting-edge configuration $P(\mathsf{C}^*(\mathbf{S}_i) \mid \mathsf{Data}_{do(\mathbf{s}_i)})$ converges to 1 with high probability. More precisely, we have the following high probability lower bound on the posterior probability of the true cutting-edge configuration.*

$$P(\mathsf{C}^*(\mathbf{S}_i) \mid \mathsf{Data}_{do(\mathbf{s}_i)}) \geq 1 - \frac{1}{1 + \alpha_1 \exp\left(\mathcal{O}(m_{\mathbf{s}_i}) - \alpha_2 \mathcal{O}\left(\sqrt{m_{\mathbf{s}_i} \ln \frac{1}{\delta}}\right)\right)} \quad w.p. \text{ at least } 1 - \delta$$

$$(4)$$

*Where $\alpha_1$ and $\alpha_2$ are constants depending on the prior and the problem instance. Thus, for any small choice of the probability $\delta$, with a sufficiently large number of samples $m_{\mathbf{s}_i}$, the posterior of the true cutting-edge configuration $P(\mathsf{C}^*(\mathbf{S}_i) \mid \mathsf{Data}_{do(\mathbf{s}_i)})$, converges to 1 with a probability at least $1 - \delta$.*

Note that the randomness in the high-probability result in Equation 4 arises from the finite sample size. Lemma 2 guarantees that with a sufficiently large number of samples from interventions on $\mathbf{S}_i$, the posterior of the true cutting edge configuration $P(\mathsf{C}^*(\mathbf{S}_i) \mid \mathsf{Data}_{do(\mathbf{s}_i)})$ converges to 1 with high probability. However, in scenarios where we want to know ahead of time how many interventional samples we need from every target $\mathbf{S}_i \in \mathcal{S}$ to ensure that the posterior of the corresponding true cutting orientations $P(\mathsf{C}^*(\mathbf{S}_i) \mid \mathsf{Data}_{do(\mathbf{s}_i)}) \geq 1 - \gamma$ with high probability, we need to determine the required number of samples. The parameter $\gamma \geq 0$ represents the error tolerance. We extend Lemma 2 to provide the minimum number of samples $m_{\mathbf{s}_i}$ required for an intervention on the target set $\mathbf{S}_i$ to ensure that the posterior of the corresponding true cutting orientations is greater than some threshold.

**Theorem 3.** *Given that the Assumption1 hold, consider an intervention target $\mathbf{S}_i \in \mathcal{S}$ such that $|\mathbf{S}_i| \leq k$ and the corresponding true cutting edge configuration $\mathsf{C}^*(\mathbf{S}_i)$. If the number of samples $m_{\mathbf{s}i}$ in $\mathsf{Data}_{do(\mathbf{s}_i)} = \{\mathbf{v}_1, \mathbf{v}_2, ..., \mathbf{v}_{m_{\mathbf{s}_i}}\}$ satisfies the following:*

$$m_{\mathbf{s}_i} = \frac{2\beta^2}{(D^{\mathbf{s}_i})^2} \ln \frac{2^{(k+1)d_m}}{\delta} + \frac{2}{D^{\mathbf{s}_i}} \ln \frac{2^{kd_m}(1-\gamma)(1-p^*)}{p^*\gamma} \tag{5}$$

*where $p^*$ is the prior assigned to the true cutting-edge configuration $\mathsf{C}^*(\mathbf{S}_i)$, then we have $P(\mathsf{C}^*(\mathbf{S}_i) \mid \mathsf{Data}_{do(\mathbf{s}_i)}) \geq 1 - \gamma$ with a probability at least $1 - \delta$.*

Since we have multiple intervention targets in the $(n, k)$-separating system of the form $\mathcal{S} = \{\mathbf{S_1}, \mathbf{S_2}, ..\mathbf{S_p}\}$ such that $|\mathbf{S}_i| \leq k$ for all $i \in [p]$. We can have a total of $p$ bad events one for every target set $\mathbf{S}_i \in \mathcal{S}$ where posterior of corresponding true cutting edge configuration $\mathsf{C}^*(\mathbf{S}_i)$ is not greater than the desired threshold $1 - \gamma$. Thus we extend the Theorem 3 as follows:

**Corollary 4.** *Given that Assumption1 hold, consider a separating system of the form $\mathcal{S} = \{\mathbf{S_1}, \mathbf{S_2}, ..\mathbf{S_p}\}$ such that $|\mathbf{S}_i| \leq k$ for all $i \in [p]$. If the number of samples $m_{\mathbf{s}_i}$ in $\mathsf{Data}_{do(\mathbf{s}_i)} = \{\mathbf{v}_1, \mathbf{v}_2, ..., \mathbf{v}_{m_{\mathbf{s}_i}}\}$ for every target set $\mathbf{S}_i \in \mathcal{S}$ satisfies the following:*

$$m_{\mathbf{s}_i} = \frac{2\beta^2}{(D^{\mathbf{s}_i})^2} \left( \ln \frac{2^{(k+1)d_m}}{\delta'} \right) + \frac{2}{D^{\mathbf{s}_i}} \left( \ln \frac{2^{kd_m}(1-\gamma)(1-p^*)}{p^*\gamma} \right) \tag{6}$$

*where $p^*$ is the prior assigned to the true cutting edge configuration $\mathsf{C}^*(\mathbf{S}_i)$ and $\delta' = \frac{\delta}{p}$, then we have $P(\mathsf{C}^*(\mathbf{S}_i) \mid \mathsf{Data}_{do(\mathbf{s}_i)}) \geq 1 - \gamma$ with probability at least $1 - \delta$ for every $\mathbf{S}_i \in \mathcal{S}$.*

To ensure that for every target set $\mathbf{S}_i \in \mathcal{S}$, we have the posterior probability of the corresponding true cutting-edge configuration $P(\mathsf{C}^*(\mathbf{S}_i) \mid \mathsf{Data}_{do(\mathbf{s}_i)}) \geq 1 - \gamma$, we need $m_{\mathbf{s}_i}$ samples (as in Equation 6) from every target. Therefore, for the causal discovery problem with $p$ targets in a separating system, we require a total of $p \cdot m_{\mathbf{s}_i}$ samples.

Based on the analysis, we propose Algorithm 1 as the main algorithm. We start by calculating a $(n, k)$-separating system $\mathcal{S}$ using the labeling procedure described in Appendix D for the essential graph $G = \mathcal{E}(\mathcal{D}^*)$, and them identify all valid configurations $C(\mathbf{S}), \forall \mathbf{S} \in \mathcal{S}$ using Algorithm 3. For each valid configuration, one can simply assume that their priors are uniform. In our algorithm, we assume that each possible DAG in $[\mathcal{D}^*]$ are equally likely to be the true DAG. Therefore we can calculate the prior more accurately using the MEC counting algorithm [Wienöbst et al., 2023]. Specifically, for each valid configuration $C_k(\mathbf{S})$, we find the MPDAG $\mathcal{M}_{C_k(S)}$ that matches with it by applying Meek Rules. $\mathcal{M}_{C_k(\mathbf{S})}$ is a chain graph with chordal components that can be oriented independently. Thus, the interventional MEC size of $\mathcal{M}_{C_k(S)}$ could be calculated by:

$$||[\mathcal{M}_{C_k(\mathbf{S})}]|| = \prod_{H \in CC(\mathcal{M}_{C_k(\mathbf{S})})} ||[H]|| \tag{7}$$

The prior of each configuration could then be estimated by:

$$P(C_k(\mathbf{S})) = \frac{||[\mathcal{M}_{C_k(\mathbf{S})}]||}{\sum_{C_j \in C(\mathbf{S})} ||[\mathcal{M}_{C_j(\mathbf{S})}]||} \tag{8}$$

If we are allowed to collect $N$ total interventional samples, we can randomly choose a target set from the $(n, k)$-separating system for each sample. Algorithm 3 iterates all valid configurations to calculate

---

**Algorithm 1:** Sample efficient causal discovery in Bayesian approach

---

**Data:** Input UCCG $G = (\mathbf{V}, \mathbf{E})$, $(n, k)$-separating system $\mathcal{S}$, and observational joint distribution $P_{obs}(\mathbf{V})$, prior of each configuration $P(C(\mathbf{S}))$, number of samples $N$

**Result:** Output the posterior $P_{\mathbf{S}}^{C(\mathbf{S})}(C(\mathbf{S})|\mathbf{v}_{do(\mathbf{S})})$ of each configuration to be true and a candidate DAG $\mathcal{D}$

**for** $i \in [N]$ **do**
  Randomly choose a target set $\mathbf{S} \in \mathcal{S}$;
  Sample $\mathbf{v}_{do(\mathbf{S})}$ from $P_{\mathbf{S}}$;
  **for** $C_k(\mathbf{S}) \in C(\mathbf{S})$ **do**
    Calculate likelihood $P(\mathbf{v}_{do(\mathbf{S})}|C_k(\mathbf{S}))$ using Algorithm 3;

  **for** $C_k(\mathbf{S}) \in C(\mathbf{S})$ **do**
    Update posterior and prior ;
    $P_{\mathbf{S}}^{C_k(\mathbf{S})}(C_k(\mathbf{S})|\mathbf{v}_{do(\mathbf{S})}) \leftarrow \frac{P(C_k(\mathbf{S})) \times P(\mathbf{v}_{do(\mathbf{S})}|C_k(\mathbf{S}))}{\sum_{C_k(\mathbf{S}) \in C(\mathbf{S})} P(C_k(\mathbf{S})) \times P(\mathbf{v}_{do(\mathbf{S})}|C_k(\mathbf{S}))}$;
    $P(C_k(\mathbf{S})) \leftarrow P_{\mathbf{S}}^{C_k(\mathbf{S})}(C_k(\mathbf{S})|\mathbf{v}_{do(\mathbf{S})})$

$\mathcal{D} \leftarrow G, \mathcal{S}_{visit} = \emptyset$;
**while** $\mathcal{S} \neq \mathcal{S}_{visit}$ **do**
  $C_k(\mathbf{S}) \leftarrow \arg\max_{C_j(\mathbf{S}) \in C(\mathbf{S}), \mathbf{S} \in \mathcal{S}} P_{\mathbf{S}}^{C_k(\mathbf{S})}(C_k(\mathbf{S})|\mathbf{v}_{do(\mathbf{S})})$;
  **if** $C_k(\mathbf{S})$ *not compatible with* $\mathcal{S}_{visit}$ **then**
    Remove $C_k(\mathbf{S})$ from $C(\mathbf{S})$;
    Pass;

  Replace edges in $\mathcal{D}$ with arcs in $C_k(\mathbf{S})$;
  $\mathcal{S}_{visit} \leftarrow \mathcal{S}_{visit} \cup \{\mathbf{S}\}$;

**return** $P_{\mathbf{S}}^{C(\mathbf{S})}(C(\mathbf{S})|\mathbf{v}_{do(\mathbf{S})}), \mathcal{D}$

---

the likelihood $P(\mathbf{v}_{do(S)}|C_k(S))$ by sampling DAG from the interventional MEC. After calculating all the likelihoods, we can update the posteriors and priors. For the last step, we need to combine all the configurations of each $(n, k)$-separating set to return a DAG. We consider a greedy approach. At each step, we choose the configuration with the highest posterior across all unvisited target sets that is compatible with chosen configurations. Given the anytime nature of our algorithm, we can propose a candidate DAG after each interventional sample. With a large enough sample number, according to Theorem 3, our algorithm will return the true DAG with a high probability.

## 6   Case Study: Estimating Causal Effects of Non-Intervenable Vertices

Here we show how to use DAG sampler to estimate the causal effect when some vertex cannot be intervened in the causal graph. The detailed problem setup is as follow. $\mathcal{D} = (\mathbf{V}, \mathbf{E})$ is an undirected underlying causal graph, $X, Y \in \mathbf{V}$ are non-adjacent vertices. We want to estimate $p(y|do(x))$ using interventional data while $X$ is not intervenable in the graph. Thus we cannot directly use the method in Perkovic [2020] to estimate the causal effect. With the DAG sampler, we can first intervene on $X$'s neighborhood $Ne(X)$ and then sample DAG to estimate the likelihood of each configuration being the true one in the causal graph. When we have large sample size, the posterior of the true configuration will go to 1 with a high probability according to Lemma 2. Then, we iterate through each configuration and their posterior to calculate the average divergence $\overline{D}$ between the estimated and true causal effect. The divergence $D$ here can be KL divergence or Total Variation Distance (TVD). If the ground truth causal effect is $p^*$, the interventional data is $\mathsf{Data}_{do(Ne(X))}$, then $\overline{D}$ is formally defined as:

$$\overline{D} = \sum_{C_i \in C(Ne(X))} D(p^* || P_{Ne(X)}^{C_i}(y|do(x))) \cdot P(C_i | \mathsf{Data}_{do(Ne(X))}) \qquad (9)$$

In this experiment, we randomly create 50 causal graphs with $n = 5, 6, 7$, $\rho = 0.3, 0.6$ using a similar approach described in Section 7. Then we randomly choose a pair of non-adjacent vertices $X, Y$. We then intervene on $X$ and collect 100,000 samples. We plot $\overline{D}_{KL}$ and $\overline{D}_{TVD}$ between estimated causal effect and the ground truth given different number of interventional samples. The mean and standard deviation are plotted in Figure 1. We can see that as the number of interventional samples increases, the estimated causal effect gets close to the ground truth and the variance is also decreasing. The difference decreases sharply w.r.t. the number of samples, showing the efficiency of our proposed approach.

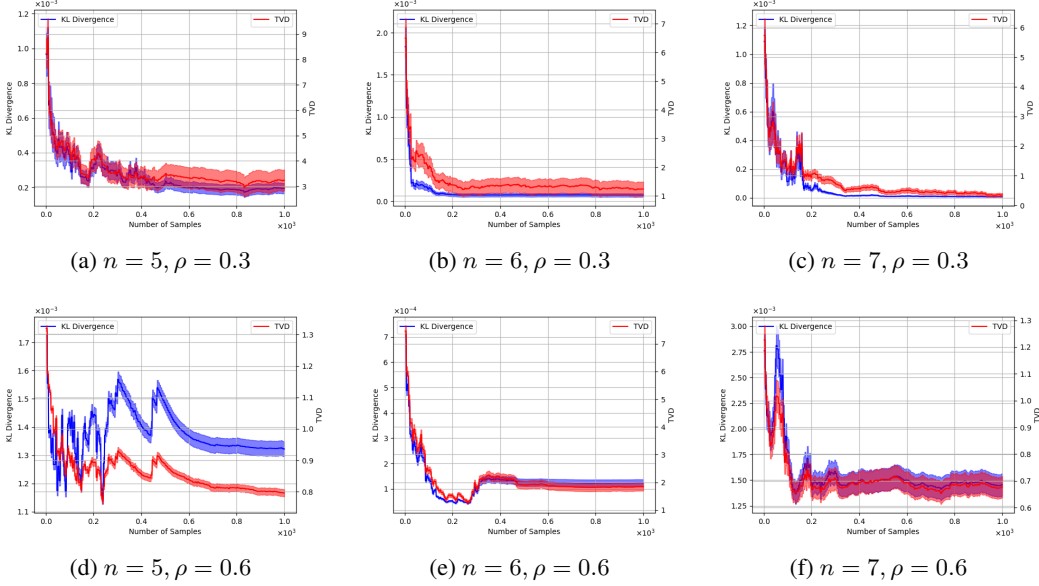

Figure 1: Average KL divergence and TVD between estimated causal effect and ground truth vs number of interventional samples for random causal graphs.

## 7 Experiments

We compare the proposed Bayesian Causal Discovery algorithm with 3 existing baselines. The first baseline, Random Intervention, intervenes randomly on the graph separating system. In each step, one interventional sample is collected, and then we perform independence tests to learn the cuts at the targets based on the collected samples from each intervention target. The second baseline is Active Structure Learning of Causal DAGs via Directed Clique Trees (DCTs) [Squires et al., 2020]. The third baseline is the adaptivity-sensitive search algorithm proposed in Choo and Shiragur [2023]. It chooses the intervention target based on the clique tree representation of a chordal graph.

We generate random connected moral DAGs with order $n$ and density $\rho$ using a modified Erdős–Rényi sample approach, similar to the process in Squires et al. [2020]. We start by generating a random ordering $\tau$ over vertices. Then, for the $i^{th}$ vertex, we sample its in-degree as $X_i = max(1, Bin(n - 1, \rho))$. The vertices precede it in the ordering are uniformly assgined as its parents. In the last step, we apply the elimination algorithm in Koller and Friedman [2009] to chordalize the graph. The elimination algorithm uses an ordering that is the reverse of $\tau$. Based on the generated graph, we randomly sample the conditional probability table (CPT) that is consistent with the graph and strictly positive. The essential graph of the generated DAG is then fed into the causal discovery algorithms together with the Bayes network that is consistent with the DAG.

To measure the performance of the algorithms, we plot the Structural Hamming Distance (SHD) between the ground truth and learned DAGs with respect to the number of collected interventional samples. SHD counts the number of edge insertions, deletions, or flips to transform from one DAG into another. For each setting of $n$ and $\rho$, we sample 50 random DAGs and calculate the mean and standard deviation of SHD by each causal discovery algorithm. The mean is plotted as a curve and the shaded area around the curve represents the standard deviation. We use Chi-Square independence test from the Causal Discovery Toolbox [Kalainathan et al., 2020] to perform statistical tests with limited samples.

The results in Figure 2 and 3 show that our algorithm outperforms other causal discovery algorithms. Figure 2 shows the performance of causal discovery algorithms on complete graphs with order 5, 6, and 7 respectively. Our algorithm uses significantly less samples to reach a low SHD and the number of samples required does not increase much with the order of the graph compared to the other algorithms. Figure 3 shows the results of graphs with the same order but varying densities. Similarly, our algorithm uses comparable number of samples across different densities. In contrast, the baseline algorithms require much more samples in denser graphs to reach the same SHD. To wrap up, in both cases, our algorithm reaches a low SHD using much fewer samples than the baselines

and is more stable across different settings. The results on larger graphs ($n = 20$) are provided in Appendix H.1. The algorithm code is provided at `https://github.com/CausalML-Lab/Bayesian_SampleEfficient_Discovery`.

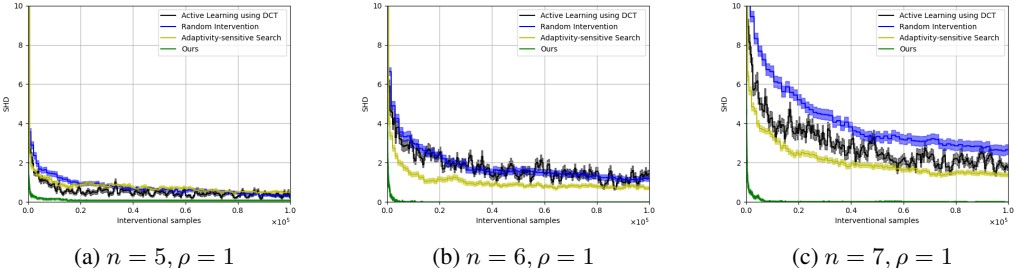

(a) $n = 5, \rho = 1$   (b) $n = 6, \rho = 1$   (c) $n = 7, \rho = 1$

Figure 2: SHD vs number of interventional samples for random complete graphs

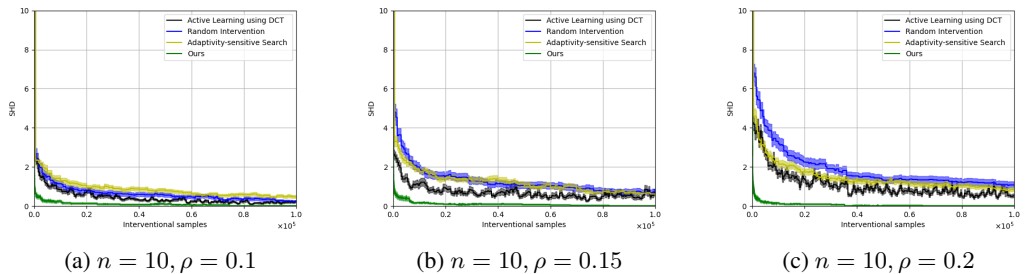

(a) $n = 10, \rho = 0.1$   (b) $n = 10, \rho = 0.15$   (c) $n = 10, \rho = 0.2$

Figure 3: SHD vs number of interventional samples for random sparse chordal graphs

## 8  Conclusion

In this work, we discuss the problem of learning causal graphs with minimum number of interventional samples. We propose an algorithm that solves this problem in a Bayesian approach. Specifically, we keep track of the posteriors of interventional distributions of each target set in the $(n, k)$-separating system and then combine the configurations that have high posteriors to propose a DAG. We show in theory that given enough samples, our algorithm can return the true causal graph with a high probability. Also, according to the Bayesian nature of our algorithm, we can stop the algorithm anytime and check the output graph. Experiments on simulated datasets show that compared with the baselines, our algorithm use significantly fewer interventional samples to achieve the same SHD. Additionally in the case study, we demonstrate that we can modify our algorithm to answer special causal queries with DAG sampling. For future extensions, it is of great interest to decrease the space of posteriors being tracked, since on large dense graphs, it will be intractable to track all interventional distributions of a target set. Besides, we can try to remove/weaken the assumptions like causal sufficiency and faithfulness while maintaining theoretical guarantees. Our work can potentially have some societal consequences including ethical considerations related to performing interventions and the risk of biased or partial understandings which may cause misled decision-making in real-world cases.

## Acknowledgement

This research has been supported in part by NSF CAREER 2239375, IIS 2348717, Amazon Research Award and Adobe Research.

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

## Supplemental Material

## A   Proof of Lemma 2:

Consider an intervention target $\mathbf{S}_i$ and a collection of all possible cutting edge configurations $\mathsf{C}_j(\mathbf{S}_i)$ for all possible cutting configurations at $\mathbf{S}_i$, i.e., for all $j \in [n_{\mathbf{S}_i}]$ and the interventional dataset of i.i.d. samples $\mathsf{Data}_{do(\mathbf{s}_i)} = \{\mathbf{v}_1, \mathbf{v}_2, ..., \mathbf{v}_{m_{\mathbf{s}_i}}\}$. We revisit some notations from the main paper. We denote the interventional distribution with cut configuration $\mathsf{C}_j(\mathbf{S}_i)$ by $P_{\mathbf{s}_i}^{\mathsf{C}_j(\mathbf{S}_i)}$. we have $n_{\mathbf{S}_i}$ different cut configurations which we denote by $\mathsf{C}_1(\mathbf{S}_i), \mathsf{C}_2(\mathbf{S}_i) .... \mathsf{C}_{n_{\mathbf{S}_i}}(\mathbf{S})$. Under faithfulness assumption we have $P_{\mathbf{s}_i}^{\mathsf{C}_a(\mathbf{S}_i)} \neq P_{\mathbf{s}_i}^{\mathsf{C}_b(\mathbf{S}_i)}$ for all $a \neq b$, $a, b \in [n_{\mathbf{S}_i}]$. Also note that for with $(n, k)$ separating system we have $n_{\mathbf{S}_i} \leq 2^{kd_m}$. Without loss of generality assume $\mathsf{C}_1(\mathbf{S}_i)$ is the true cutting edge configuration. We use the notation $D_{ab}^{\mathbf{s}_i} := D_{KL}(P_{\mathbf{s}_i}^{\mathsf{C}_a(\mathbf{S}_i)}(\mathbf{V}) \mid\mid P_{\mathbf{s}_i}^{\mathsf{C}_b(\mathbf{S}_i)}(\mathbf{V})) > 0 \ \forall a \neq b$, $\forall a, b \in [n_{\mathbf{S}_i}]$ and $D^{\mathbf{s}_i} := \min_{\forall a \neq b, \ \forall a, b \in [n_{\mathbf{S}_i}]} D_{ab}^{\mathbf{s}_i}$.

Let us consider $P_{\mathbf{s}_i}^{\mathsf{C}_1(\mathbf{S}_i)}$ and $P_{\mathbf{s}_i}^{\mathsf{C}_2(\mathbf{S}_i)}$ only for now.

$$Let \ -L = -\log \frac{P_{\mathbf{s}_i}^{\mathsf{C}_2(\mathbf{S}_i)}(\mathbf{v}_1, \mathbf{v}_2, ..., \mathbf{v}_{m_{\mathbf{s}_i}})}{P_{\mathbf{s}_i}^{\mathsf{C}_1(\mathbf{S}_i)}(\mathbf{v}_1, \mathbf{v}_2, ..., \mathbf{v}_{m_{\mathbf{s}_i}})} \tag{10}$$

Since interventional dataset is composed of i.i.d. samples we have:

$$-L = -\sum_{j=1}^{m_{\mathbf{s}_i}} \log \frac{P_{\mathbf{s}_i}^{\mathsf{C}_2(\mathbf{S}_i)}(\mathbf{v}_j)}{P_{\mathbf{s}_i}^{\mathsf{C}_1(\mathbf{S}_i)}(\mathbf{v}_j)} \tag{11}$$

$$Let \ L_j = -\log \frac{P_{\mathbf{s}_i}^{\mathsf{C}_2(\mathbf{S}_i)}(\mathbf{v}_j)}{P_{\mathbf{s}_i}^{\mathsf{C}_1(\mathbf{S}_i)}(\mathbf{v}_j)} \tag{12}$$

$$-L = \sum_{j=1}^{m_{\mathbf{s}_i}} L_j \tag{13}$$

$$\frac{1}{n} \sum_{j=1}^{m_{\mathbf{s}_i}} L_j = -\frac{L}{n} \tag{14}$$

Since $\mathsf{C}_1(\mathbf{S}_i)$ is the true cutting edge configuration so the interventional samples are actually sampled from $P_{\mathbf{s}_i}^{\mathsf{C}_1(\mathbf{S}_i)}$. So we have the following:

$$\mathbb{E}[L_j] = \mathbb{E}_{\mathbf{v}_j \sim P_{\mathbf{s}_i}^{\mathsf{C}_1(\mathbf{S}_i)}} \left[ -\log \frac{P_{\mathbf{s}_i}^{\mathsf{C}_2(\mathbf{S}_i)}(\mathbf{v}_j)}{P_{\mathbf{s}_i}^{\mathsf{C}_1(\mathbf{S}_i)}(\mathbf{v}_j)} \right] \tag{15}$$

$$\mathbb{E}[L_j] = \mathbb{E}_{\mathbf{v}_j \sim P_{\mathbf{s}_i}^{\mathsf{C}_1(\mathbf{S}_i)}} \left[ \log \frac{P_{\mathbf{s}_i}^{\mathsf{C}_1(\mathbf{S}_i)}(\mathbf{v}_j)}{P_{\mathbf{s}_i}^{\mathsf{C}_2(\mathbf{S}_i)}(\mathbf{v}_j)} \right] \tag{16}$$

$$\mathbb{E}[L_j] = D_{KL}(P_{\mathbf{s}_i}^{\mathsf{C}_1(\mathbf{S}_i)}(\mathbf{V}) \mid\mid P_{\mathbf{s}_i}^{\mathsf{C}_2(\mathbf{S}_i)}(\mathbf{V})) = D_{12}^{\mathbf{s}_i} > 0 \tag{17}$$

Under the Assumption $\mid L_j \mid \leq \beta \ \forall j \in [m_{\mathbf{s}_i}]$ for all $L_j = -\log \frac{P_{\mathbf{s}_i}^{\mathsf{C}_2(\mathbf{S}_i)}(\mathbf{v}_j)}{P_{\mathbf{s}_i}^{\mathsf{C}_1(\mathbf{S}_i)}(\mathbf{v}_j)}$. using hoeffding inequality with $m_{\mathbf{s}_i}$ samples we have:

$$P(\mid -L - m_{\mathbf{s}_i} D_{12}^{\mathbf{s}_i} \mid \geq \epsilon) \leq 2e^{\frac{-2\epsilon^2}{4\beta^2 m_{\mathbf{s}_i}}} \tag{18}$$

Suppose $\frac{\delta}{2^{kd_m}} = 2e^{\frac{-2\epsilon^2}{4\beta^2 m_{\mathbf{s}_i}}}$ which implies $\epsilon = \sqrt{2\beta^2 m_{\mathbf{s}_i} \ln \frac{2}{\frac{\delta}{2^{kd_m}}}} = \sqrt{2\beta^2 m_{\mathbf{s}_i} \ln \frac{2 \times 2^{kd_m}}{\delta}}$

Thus we have the following:

$$P(\ |-L - m_{\mathbf{s}_i} D^{\mathbf{s}_i}\ | \ \geq \sqrt{2\beta^2 m_{\mathbf{s}_i} \ln \frac{2 \times 2^{kd_m}}{\delta}}) \leq \frac{\delta}{2^{kd_m}} \tag{19}$$

which implies that with probability of at least $1 - \frac{\delta}{2^{kd_m}}$ we have:

$$-L \geq m_{\mathbf{s}_i} D_{12}^{\mathbf{s}_i} - \sqrt{2\beta^2 m_{\mathbf{s}_i} \ln \frac{2 \times 2^{kd_m}}{\delta}} \tag{20}$$

Using the definition of $L$ we have:

$$-\log \frac{P_{\mathbf{s}_i}^{\mathsf{C}_2(\mathbf{S}_i)}(\mathbf{v}_1, \mathbf{v}_2, ..., \mathbf{v}_{m_{\mathbf{s}_i}})}{P_{\mathbf{s}_i}^{\mathsf{C}_1(\mathbf{S}_i)}(\mathbf{v}_1, \mathbf{v}_2, ..., \mathbf{v}_{m_{\mathbf{s}_i}})} \geq m_{\mathbf{s}_i} D_{12}^{\mathbf{s}_i} - \sqrt{2\beta^2 m_{\mathbf{s}_i} \ln \frac{2 \times 2^{kd_m}}{\delta}} \quad (w.p. \ at \ least \quad 1 - \frac{\delta}{2^{kd_m}}) \tag{21}$$

$$\frac{P_{\mathbf{s}_i}^{\mathsf{C}_2(\mathbf{S}_i)}(\mathbf{v}_1, \mathbf{v}_2, ..., \mathbf{v}_{m_{\mathbf{s}_i}})}{P_{\mathbf{s}_i}^{\mathsf{C}_1(\mathbf{S}_i)}(\mathbf{v}_1, \mathbf{v}_2, ..., \mathbf{v}_{m_{\mathbf{s}_i}})} \leq e^{-\left(m_{\mathbf{s}_i} D_{12}^{\mathbf{s}_i} - \sqrt{2\beta^2 m_{\mathbf{s}_i} \ln \frac{2 \times 2^{kd_m}}{\delta}}\right)} \quad (w.p. \ at \ least \quad 1 - \frac{\delta}{2^{kd_m}}) \tag{22}$$

$$P_{\mathbf{s}_i}^{\mathsf{C}_2(\mathbf{S}_i)}(\mathbf{v}_1, \mathbf{v}_2, ..., \mathbf{v}_{m_{\mathbf{s}_i}}) \leq e^{-\left(m_{\mathbf{s}_i} D_{12}^{\mathbf{s}_i} - \sqrt{2\beta^2 m_{\mathbf{s}_i} \ln \frac{2 \times 2^{kd_m}}{\delta}}\right)} P_{\mathbf{s}_i}^{\mathsf{C}_1(\mathbf{S}_i)}(\mathbf{v}_1, \mathbf{v}_2, ..., \mathbf{v}_{m_{\mathbf{s}_i}}) \tag{23}$$

The equation 23 holds with probability at least $1 - \frac{\delta}{2^{kd_m}}$. Similarly using the union bound we have with probability at least $1 - \delta$ we have the following $\forall j \neq 1$ , $j \in [n_{\mathbf{S}_i}]$:

$$P_{\mathbf{s}_i}^{\mathsf{C}_j(\mathbf{S}_i)}(\mathbf{v}_1, \mathbf{v}_2, ..., \mathbf{v}_{m_{\mathbf{s}_i}}) \leq e^{-\left(m_{\mathbf{s}_i} D_{1j}^{\mathbf{s}_i} - \sqrt{2\beta^2 m_{\mathbf{s}_i} \ln \frac{2 \times 2^{kd_m}}{\delta}}\right)} P_{\mathbf{s}_i}^{\mathsf{C}_1(\mathbf{S}_i)}(\mathbf{v}_1, \mathbf{v}_2, ..., \mathbf{v}_{m_{\mathbf{s}_i}}) \tag{24}$$

Now, suppose we assign a non-zero prior $p_j > 0$ to the all cut configuration $\mathsf{C}_j(\mathbf{S}_i) \ \forall j \in [n_{\mathbf{S}_I}]$ such that $\sum_{j=1}^{n} p_i = 1$. The posterior distribution can be written as:

$$P(\mathsf{C}_1(\mathbf{S}_i) \mid \mathsf{Data}_{do(\mathbf{s}_i)}) = \frac{P_{\mathbf{s}_i}^{\mathsf{C}_1(\mathbf{S}_i)}(\mathbf{v}_1, \mathbf{v}_2, ..., \mathbf{v}_{m_{\mathbf{s}_i}}) \, p_1}{\sum_{j=1}^{n_{\mathbf{S}_i}} P_{\mathbf{s}_i}^{\mathsf{C}_j(\mathbf{S}_i)}(\mathbf{v}_1, \mathbf{v}_2, ..., \mathbf{v}_{m_{\mathbf{s}_i}}) \, p_j} \tag{25}$$

From the result in Equation 24 with probability at least $1 - \delta$ we have:

$$P(\mathsf{C}_1(\mathbf{S}_i) \mid \mathsf{Data}_{do(\mathbf{s}_i)}) \geq \frac{P_{\mathbf{s}_i}^{\mathsf{C}_1(\mathbf{S}_i)}(\mathbf{v}_1, \mathbf{v}_2, ..., \mathbf{v}_{m_{\mathbf{s}_i}}) \, p_1}{P_{\mathbf{s}_i}^{\mathsf{C}_1(\mathbf{S}_i)}(\mathbf{v}_1, \mathbf{v}_2, ..., \mathbf{v}_{m_{\mathbf{s}_i}}) \, p_1 + \sum_{j=2}^{n_{\mathbf{S}_i}} P_{\mathbf{s}_i}^{\mathsf{C}_1(\mathbf{S}_i)}(\mathbf{v}_1, \mathbf{v}_2, ..., \mathbf{v}_{m_{\mathbf{s}_i}}) \, e^{-\left(m_{\mathbf{s}_i} D_{1j}^{\mathbf{s}_i} - \sqrt{2\beta^2 m_{\mathbf{s}_i} \ln \frac{2 \times 2^{kd_m}}{\delta}}\right)} p_j} \tag{26}$$

With probability at least $1 - \delta$ we have:

$$P(\mathsf{C}_1(\mathbf{S}_i) \mid \mathsf{Data}_{do(\mathbf{s}_i)}) \geq \frac{p_1}{p_1 + \sum_{j=2}^{n_{\mathbf{S}_i}} e^{-\left(m_{\mathbf{s}_i} D_{1j}^{\mathbf{s}_i} - \sqrt{2\beta^2 m_{\mathbf{s}_i} \ln \frac{2 \times 2^{kd_m}}{\delta}}\right)} p_j} \tag{27}$$

Since $n_{\mathbf{S}_i} \leq 2^{kd_m}$ and $D^{\mathbf{s}_i} := \min_{\forall a \neq b \, , \, \forall a,b \in [n_{\mathbf{S}_i}]} D_{ab}^{\mathbf{s}_i}$. With probability at least $1 - \delta$ we have:

$$P(\mathsf{C}_1(\mathbf{S}_i) \mid \mathsf{Data}_{do(\mathbf{s}_i)}) \geq \frac{p_1}{p_1 + (1 - p_1) 2^{kd_m} e^{-\left(m_{\mathbf{s}_i} D^{\mathbf{s}_i} - \sqrt{2\beta^2 m_{\mathbf{s}_i} \ln \frac{2 \times 2^{kd_m}}{\delta}}\right)}} \tag{28}$$

$$P(\mathsf{C}_1(\mathbf{S}_i) \mid \mathsf{Data}_{do(\mathbf{s}_i)}) \geq 1 - \frac{1}{1 + \frac{p_1}{(1-p_1)2^{kd_m}} e^{\left(m_{\mathbf{s}_i} D^{\mathbf{s}_i} - \sqrt{2\beta^2 m_{\mathbf{s}_i} \ln \frac{2 \times 2^{kd_m}}{\delta}}\right)}} \quad (w.p. \ at \ least \ 1 - \delta)$$

(29)

which can equivalently written as follows:

$$P(\mathsf{C}^*(\mathbf{S}_i) \mid \mathsf{Data}_{do(\mathbf{s}_i)}) \geq 1 - \frac{1}{1 + \alpha_1 \exp\left(\mathcal{O}(m_{\mathbf{s}_i}) - \alpha_2 \mathcal{O}\left(\sqrt{m_{\mathbf{s}_i} \ln \frac{1}{\delta}}\right)\right)} \quad (w.p. \ at \ least \ 1 - \delta)$$

(30)

Where $\alpha_1$ and $\alpha_2$ are constants depending on the priors used and the problem instance. Note the in our proof the true cutting edge configuration $\mathsf{C}^*(\mathbf{S}_i) = \mathsf{C}_1(\mathbf{S}_i)$. This completes the proof of Lemma 2.

## B    Proof of Theorem 3:

The Proof of the Theorem 3 is the continuation of the proof of Lemma 2. Suppose we want $P(\mathsf{C}_1(\mathbf{S}_i) \mid \mathsf{Data}_{do(\mathbf{s}_i)}) \geq 1 - \gamma$ we have:

$$(1 - p_1)2^{kd_m} e^{-\left(m_{\mathbf{s}_i} D^{\mathbf{s}_i} - \sqrt{2\beta^2 m_{\mathbf{s}_i} \ln \frac{2 \times 2^{kd_m}}{\delta}}\right)} \leq \frac{p_1 \gamma}{1 - \gamma}$$

(31)

$$e^{-\left(m_{\mathbf{s}_i} D^{\mathbf{s}_i} - \sqrt{2\beta^2 m_{\mathbf{s}_i} \ln \frac{2 \times 2^{kd_m}}{\delta}}\right)} \leq \frac{p_1 \gamma}{2^{kd_m}(1 - \gamma)(1 - p_1)}$$

(32)

$$m_{\mathbf{s}_i} D^{\mathbf{s}_i} - \sqrt{2\beta^2 m_{\mathbf{s}_i} \ln \frac{2 \times 2^{kd_m}}{\delta}} \geq \log \frac{2^{kd_m}(1 - \gamma)(1 - p_1)}{p_1 \gamma}$$

(33)

Solving the above equation for number of interventional samples $m_{\mathbf{s}_i}$ we have:

$$m_{\mathbf{s}_i} \geq \frac{4\beta^2 \ln \frac{2^{(k+1)d_m}}{\delta} + 4D^{\mathbf{s}_i} \ln \frac{2^{kd_m}(1-\gamma)(1-p_1)}{p_1\gamma} + \sqrt{16\beta^4 \ln^2 \frac{2^{(k+1)d_m}}{\delta} + 32\beta^2 D^{\mathbf{s}_i} \ln \frac{2^{(k+1)d_m}}{\delta} \ln \frac{2^{kd_m}(1-\gamma)(1-p_1)}{p_1\gamma}}}{4(D^{\mathbf{s}_i})^2}$$

(34)

In order to simplify the expression we select the number of interventional samples $m_{\mathbf{s}_i}$ as follows:

$$m_{\mathbf{s}_i} = \frac{4\beta^2 \ln \frac{2^{(k+1)d_m}}{\delta} + 4D^{\mathbf{s}_i} \ln \frac{2^{kd_m}(1-\gamma)(1-p_1)}{p_1\gamma} + \sqrt{\left(4\beta^2 \ln \frac{2^{(k+1)d_m}}{\delta} + 4D^{\mathbf{s}_i} \ln \frac{2^{kd_m}(1-\gamma)(1-p_1)}{p_1\gamma}\right)^2}}{4(D^{\mathbf{s}_i})^2}$$

(35)

$$m_{\mathbf{s}_i} = \frac{4\beta^2 \ln \frac{2^{(k+1)d_m}}{\delta} + 4D^{\mathbf{s}_i} \ln \frac{2^{kd_m}(1-\gamma)(1-p_1)}{p_1\gamma} +}{2(D^{\mathbf{s}_i})^2}$$

(36)

$$m_{\mathbf{s}_i} = \frac{2\beta^2}{(D^{\mathbf{s}_i})^2} \ln \frac{2^{(k+1)d_m}}{\delta} + \frac{2}{D^{\mathbf{s}_i}} \ln \frac{2^{kd_m}(1-\gamma)(1-p_1)}{p_1\gamma}$$

(37)

for the above choice of number of samples $m_{\mathbf{s}_i}$ we have the following result:

$$P(\mathsf{C}_1(\mathbf{S}_i) \mid \mathsf{Data}_{do(\mathbf{s}_i)}) \geq 1 - \gamma \quad (w.p. \ at \ least \quad 1 - \delta)$$

(38)

Now, suppose $p^*$ is the prior assigned to the true cutting-edge configuration $C^*(\mathbf{S}_i)$. Consequently, for the following choice of number of samples $m_{\mathbf{s}_i}$:

$$m_{\mathbf{s}_i} = \frac{2\beta^2}{(D^{\mathbf{s}_i})^2} \ln \frac{2^{(k+1)d_m}}{\delta} + \frac{2}{D^{\mathbf{s}_i}} \ln \frac{2^{kd_m}(1-\gamma)(1-p^*)}{p^*\gamma} \tag{39}$$

We have $P(C^*(\mathbf{S}_i) \mid \mathrm{Data}_{do(\mathbf{s}_i)}) \geq 1 - \gamma$ with probability at least $1 - \delta$. Note the in our proof the true cutting edge configuration $C^*(\mathbf{S}_i) = C_1(\mathbf{S}_i)$. This completes the proof of Theorem 3.

## C   Proof of the Corollary 4:

Consider a separating system of the form $\mathcal{S} = \{\mathbf{S_1}, \mathbf{S_2}, ..\mathbf{S_p}\}$ such that $|\mathbf{s}_i| \leq k$ for all $i \in [p]$. The Proof of the Corollary is simple application of union to result in Theorem 3. if replace $\delta$ by $\frac{\delta}{p}$ in the Equation 39 for number of samples $m_{\mathbf{s}_i}$ then for a particular $\mathbf{S}_i \in \mathcal{S}$ we have $P(C^*(\mathbf{S}_i) \mid \mathrm{Data}_{do(\mathbf{s}_i)}) \geq 1 - \gamma$ with probability at least $1 - \frac{\delta}{p}$. Taking union bound we have the following result:

If the number of samples $m_{\mathbf{s}_i}$ in $\mathrm{Data}_{do(\mathbf{s}_i)} = \{\mathbf{v}_1, \mathbf{v}_2, ..., \mathbf{v}_{m_{\mathbf{s}_i}}\}$ for every target set $\mathbf{S}_i \in \mathcal{S}$ satisfies the following:

$$m_{\mathbf{s}_i} = \frac{2\beta^2}{(D^{\mathbf{s}_i})^2} \ln \frac{2^{(k+1)d_m}}{\delta'} + \frac{2}{D^{\mathbf{s}_i}} \ln \frac{2^{kd_m}(1-\gamma)(1-p^*)}{p^*\gamma} \tag{40}$$

where $p^*$ is the prior assigned to the true cutting edge configuration $C^*(\mathbf{S}_i)$ and $\delta' = \frac{\delta}{p}$. Then we have with $P(C^*(\mathbf{S}_i) \mid \mathrm{Data}_{do(\mathbf{s}_i)}) \geq 1 - \gamma$ with probability at least $1 - \delta$ for every target set $\mathbf{S}_i \in \mathcal{S}$. This completes the proof of Corollary 4.

## D   Procedure to Construct Separating Systems

### D.1   $(n, k)$-Separating System

**Lemma 5** ( [Shanmugam et al., 2015]). *There exists a labeling procedure that gives distinct labels of length $l$ for all elements in $[n]$ using letter from the integer alphabet $\{0, 1, ..., a\}$, where $l = \lceil \log_a n \rceil$. Furthermore, in every position, any integer letter is used at most $\lceil \frac{n}{a} \rceil$ times.*

The labeling method in Lemma 5 from Shanmugam et al. [2015] is described as:
**Labelling Procedure:** Let $a > 1$ be a positive integer. Let $x$ be the integer such that $a^x < n \leq a^{x+1}$. $x + 1 = \lceil \log_a n \rceil$. Every element $j \in [n]$ is given a label $L(j)$ which is a string of integers of length $x + 1$ drawn from the alphabet $\{0, 1, ..., a\}$ of size $a + 1$. Let $n = p_d a^d + r_d$ and $n = p_{d-1} a^{d-1} + r_{d-1}$ for any integers $p_d, p_{d-1}, r_d, r_{d-1}$, where $r_d < a^d$ and $r_{d-1} < a^{d-1}$. Now, we describe the sequence of the $d$-th digit across the string labels of all elements from 1 to $n$:

1. Repeat the integer 0 a total of $a^{d-1}$ times, and then repeat the subsequent integer, 1, also $a^{d-1}$ times from $\{0, 1, ..., a - 1\}$ till $p_d a^d$.

2. Following this, repeat the integer $o$ a number of times equal to $\lceil \frac{r_d}{a} \rceil$, and the repeat the integer 1 $\lceil \frac{r_d}{a} \rceil$ times, continuing this pattern until we reach the $n$th position. It is evident that the $n$th integer in the sequence will not exceed $n - 1$.

3. Each integer that appears beyond the position $a^{d-1} p_{d-1}$ is incremented by 1.

Once we have a set of $n$ string labels, we can easily construct a $(n, k)$-separating system using the following Lemma:

**Lemma 6** ( [Shanmugam et al., 2015]). *Consider an alphabet $\mathcal{A} = [0 : \lceil \frac{n}{k} \rceil]$ of size $\lceil \frac{n}{k} \rceil + 1$ where $k < \frac{n}{2}$. Label every element of an $n$ element set using a distinct string of letters from $\mathcal{A}$ of length $l = \lceil \log_{\lceil \frac{n}{k} \rceil} n \rceil$ using the labeling procedure in Lemma 5 with $a = \lceil \frac{n}{k} \rceil$. For every $1 \leq a \leq l$ and $1 \leq b \leq \lceil \frac{n}{k} \rceil$, we choose the subset $I_{a,b}$ of vertices whose string's $a$-th letter is $b$. The set of all such subsets $\mathcal{S} = \{\mathbf{s}_{a,b}\}$ is a $k$-separating system on $n$ elements and $|\mathcal{S}| \leq (\lceil \frac{n}{k} \rceil) \lceil \log_{\lceil \frac{n}{k} \rceil} n \rceil$.*

## D.2 $G$-Separating System

While an $(n, k)$-separating system separates every edge in the complete graph of order $n$, usually the causal graphs are sparse with much fewer edges. Thusly, we also provide a simple algorithm to construct a separating system that cuts all the edges in a graph.

**Definition 4** ($G$-Separating System, Definition 3 of Kocaoglu et al. [2017]). *Given an undirected graph $G = (\mathbf{V}, \mathbf{E})$, a set of subsets $\mathcal{I} \subseteq 2^{\mathbf{V}}$ is a $G$-separating system if for every edge $\{u, v\} \in \mathbf{E}$, there exists $I \in \mathcal{I}$ such that either $(u \in I_i$ and $v \notin I_i)$ or $(u \notin I_i$ and $v \in I_i)$.*

Kocaoglu et al. [2017] show that an intervention set $\mathcal{I}$ learns every DAG entailed by $G$ if and only if it is a $G$-separating system. For a graph with $n$ vertices, a $G$-separating system $\mathcal{S} = S_1, S_2, ..., S_m$, such that $|S_i| \leq k; \forall i \in [m]$, is called a $(n, k)$-separating system. There are many ways to get a $G$-separating system. Algorithm 2 gives a simple algorithm that uses vertex coloring. We first find the perfect elimination order (PEO) $L$ of $G$. PEO can be found by maximum cardinality search [Tarjan and Yannakakis, 1984]. Then based on the reverse PEO, we apply greedy coloring to assign each vertex with a number $i \in [\omega]$ where $\omega$ is the clique number of $G$. $\omega$ colors are guaranteed to color $G$ since $G$ is a perfect graph. In each step, we assign a vertex with the minimum color number that is not used by its visited neighbors, thus no adjacent vertices will have the same color. For each vertex, the union of it and its visited neighbors induce a clique, meaning that its color number is bounded by $\omega$. Consequently, Algorithm 2 returns a $G$-separating system with $\omega$ intervention sets as it cuts every edge in $G$. We use $f(\cdot)$ to represent a function that maps a set of vertices to a set of colors that are used by the vertices. $\mathcal{N}(v)$ denotes the neighborhood of a vertex $v \in \mathbf{V}$.

---
**Algorithm 2:** $G$-separating system of a given unoriented graph $G$

---
**Data:** Input UCCG $G = (\mathbf{V}, \mathbf{E}), |\mathbf{V}| = n$
**Result:** Output the $G$-separating system $\mathcal{S}$
Initialization: Find the PEO $L$ and $\omega$ of $G$ by Maximum Cardinality Search;
$L \leftarrow$ Reverse $L$, $\mathbf{S}_i = \emptyset, \forall i \in [\omega], f(\mathbf{V}) = \{0\}$;
**for** $i = 1$ *to* $n$ **do**
    $c_i \leftarrow \min\{[\omega] - f(\mathcal{N}(L_i))\}$;
    $f(L_i) \leftarrow \{c_i\}$;
    $\mathbf{S}_{c_i} \leftarrow \mathbf{S}_{c_i} \cup \{L_i\}$;
$\mathcal{S} = \cup_{i \in [\omega]} \{\mathbf{S}_i\}$;
**return** $\mathcal{S}$;

---

# E   Algorithm for Enumerating Causal Effects

---
**Algorithm 3:** Enumerating all interventional distributions and priors of an intervention set $I$ on an undirected graph $G$

---
**Data:** Input UCCG $G = (\mathbf{V}, \mathbf{E})$, intervention set $\mathbf{S} \subseteq \mathbf{V}$
**Result:** Output all possible interventional distribution $P_{\mathbf{S}}^{C(\mathbf{S})}(\mathbf{V})$ and $P(C(\mathbf{S}))$
Initialization: Enumerate all possible edge configurations $C(\mathbf{S})$ that are adjacent to vertices in $\mathbf{S}$;
**for** $C_k(\mathbf{S})$ *in* $C(\mathbf{S})$ **do**
    **if** $C_k(\mathbf{S})$ *not valid* **then**
        Remove $C_k(\mathbf{S})$ from $C(\mathbf{S})$;
        Pass;
    Get the MPDAG $\mathcal{M}_{C_k(\mathbf{S})}$ by applying Meek Rules;
    **for** $H \in CC(\mathcal{M}_{C_k(\mathbf{S})})$ **do**
        Sample a DAG $\mathcal{D}_H$ from $[H]$;
        Replace edges in $G$ with arcs in $\mathcal{D}_H$;
    Calculate $P_{\mathbf{S}}^{C_k(\mathbf{S})}(\mathbf{V})$ with Equation 1;
    Calculate the prior $P(C_k(\mathbf{S}))$ with Equation 8;
**return** $C(\mathbf{S}), P_{\mathbf{S}}^{C(\mathbf{S})}(\mathbf{V}), P(C(\mathbf{S}))$;

---

# F  Discussion on Computational Complexity

For Bayesian approach of learning causal graph, we mainly care about the space complexity because for run time, all the steps (e.g. calculating likelihood, updating posteriors) are efficient. For dense graphs, a fully Bayesian method and our algorithm are both computationally expensive. For example, for a complete graph with order $n$, Fully Bayesian method needs to store $\mathcal{O}(n!)$ distributions because there are such many DAGs in the MEC. Our method needs to track $\mathcal{O}(2^{n-1})$ posteriors, which is less than fully Bayesian but still exponentially many and would be intractable when $n$ is large. However, our method would be a lot better when the given graph is sparse, i.e. the maximum degree $d$ in the UCCG $G$ is around $\log n$. If we consider atomic interventions, the valid configurations around a vertex $\mathbf{v} \in \mathbf{V}$, $|C(\mathbf{v})| < 2^d = n$. While the MEC size of a UCCG with order $n$ is at least $n$, i.e., $|[G]| \geq n$. The equality holds if and only if $G$ is a tree. Thus, our method is at least as good as a fully Bayesian method. However, even for sparse graphs, its MEC size could be exponentially many in $n$. Let's consider the following example. Given a sparse UCCG $G$ with order $n$. $G$ consists of $\left[\frac{n}{[\log n]}\right]$ small cliques. Each small clique contains $[\log n]$ vertices. The cliques are connected together to form a 'clique line'. The MEC size of $G$ would be:

$$|[G]| \geq ([\log n]!)^{\left[\frac{n}{[\log n]}\right]} \tag{41}$$

According to Stirling's Approximation, we have:

$$|[G]| \in \mathcal{O}((\frac{[\log n]}{e})^{[\log n]}(\log n)^{-\frac{1}{2}}) \tag{42}$$

$$\in \mathcal{O}([\log n]^{[\log n]}n^{-1}[\log n]^{-\frac{1}{2}}) \tag{43}$$

$$\in \mathcal{O}(e^{\log[\log n] \cdot [\log n]}n^{-1}[\log n]^{-\frac{1}{2}}) \tag{44}$$

$$\in \mathcal{O}(n^{\log[\log n]-1}[\log n]^{-\frac{1}{2}}) \tag{45}$$

Which is approximately $\mathcal{O}(n^{\log[\log n]})$. If we consider an interventional target of size $k$ for our algorithm, there are at most $2^{kd} = n^k$ configurations to track. If $n$ is large enough, $k < \log[\log n], \forall k > 0$. Thus, the MEC size for $G$ is greater than any polynomial complexity, while our algorithm tracks at most polynomially many configurations.

# G  Further Discussion on Case Study

In Section 4.2, we mentioned that the Causal Identification Formula for MPDAG from Perkovic [2020] could also be used when enumerating the causal effects. Here we show that our use of DAG sampler provides a more flexible and general approach of causal discovery with special queries that do not rely on the cut configurations of the MPDAG. In many real-world settings, our focus is on specific queries rather than understanding the entire graph. For instance, when a cloud computing system encounters an error, we aim to efficiently identify the root cause. Similarly, when a patient exhibits symptoms of cancer, we want to determine if certain attributes of the patient are direct causes of the cancer. Here, we consider the scenario mentioned in Malinsky [2024], where the goal is to estimate the posterior probability of a set of variables being the adjustment set of a causal query given the essential graph. The adjustment set is crucial in estimating the causal effect. Given the essential graph $G$, denote the variable of a set $\mathbf{S}$ being the adjustment set as $A_{\mathbf{S}}$. $A_{\mathbf{S}} = 1$ if $\mathbf{S}$ is the adjustment set and 0 otherwise. We want to estimate $P(A_{\mathbf{S}}|\mathbf{v})$, where $\mathbf{v}$ could be any data collected, both observational and interventional, Denote the true DAG as $\mathcal{D}^*$, the posterior could be expressed as:

$$P(A_{\mathbf{S}}|\mathbf{v}) \propto \sum_{\mathcal{D} \in [\mathcal{E}(\mathcal{D}^*)]} P(A_{\mathcal{S}}|\mathbf{v}, \mathcal{D})P(\mathbf{v}|\mathcal{D})P(\mathcal{D}) \tag{46}$$

Where $P(A_{\mathcal{S}}|\mathbf{v}, \mathcal{D}) = P(A_{\mathcal{S}}|\mathcal{D})$ is an indicator function. Equation 46 could be rewritten as:

$$\mathbb{E}_{\mathcal{S}} = \mathbb{E}_{\mathcal{D} \sim [\mathcal{E}(\mathcal{D}^*)]}[P(A_{\mathcal{S}}|\mathcal{D})P(\mathbf{v}|\mathcal{D})P(\mathcal{D})] \tag{47}$$

Which can be approximated by sampling DAGs from the MEC $[\mathcal{D}^*]$. If we have $M$ DAG samples, define $\mathbb{S}$ as

$$\mathbb{S} = \sum_{i=1}^{M} P(A_{\mathcal{S}}|\mathcal{D}_i)P(\mathbf{v}|\mathcal{D}_i)P(\mathcal{D}_i); \mathcal{D}_i \in [\mathcal{E}(\mathcal{D}^*)] \tag{48}$$

According to Hoeffding Inequality, the difference between $\mathbb{S}$ and $\mathbb{E}_\mathcal{S}$ could be bounded by

$$P\left(\left|\frac{\mathbb{S}}{M} - \mathbb{E}_\mathcal{S}\right| \geq \epsilon\right) \leq \exp(-2M\epsilon^2) \tag{49}$$

By estimating the posteriors for both $P(A_\mathbf{S} = 1|\mathbf{v})$ and $P(A_\mathbf{S} = 0|\mathbf{v})$, we can normalize them to get the exact posterior. For large and dense graphs which are hard to iterate through each DAG, we could estimate efficiently by sampling DAGs according to intended accuracy. With enough DAG samples, it shows that the estimation converges to the true posterior with a high probability.

## H  Additional Experiments

### H.1  Additional Experiments on Large Graphs

We further test our proposed algorithm against the baseline methods on larger random chordal graphs. The results are shown in Figure 4. We can see that our algorithm converges very fast to a low SHD. The performance of our algorithm does not differ much from that in small graphs. The baselines, on the other hand, require a lot more samples to reach the same SHD as in small graphs.
For extremely large graphs, it would be intractable to store the whole observational distribution, instead we could all the conditional distributions of the factorizations. Given large enough observational samples, the conditional distributions will be accurate and then we can perform the Algorithm 1 for Bayesian learning.

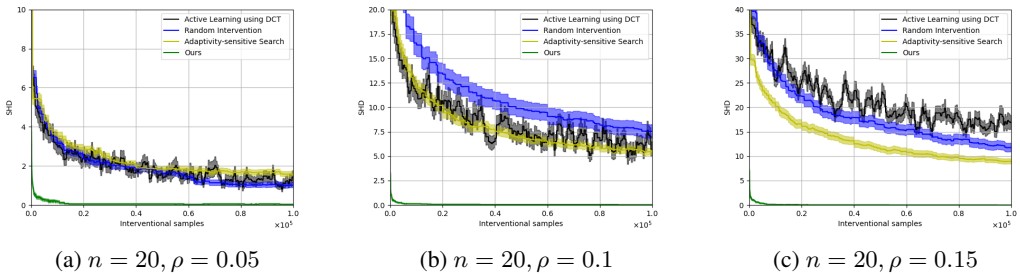

(a) $n = 20, \rho = 0.05$      (b) $n = 20, \rho = 0.1$      (c) $n = 20, \rho = 0.15$

Figure 4: SHD vs number of interventional samples for large random Erdős-Rényi chordal graphs

### H.2  Additional Experiment Scale-Free Graphs

In this experiment, we demonstrate the performance of our proposed algorithm on scale-free graphs. For each experiment settings, we generate 50 random DAGs from Barabási-Albert (BA) model. The results are plotted in Figure 5. Our proposed method still outperforms other methods.

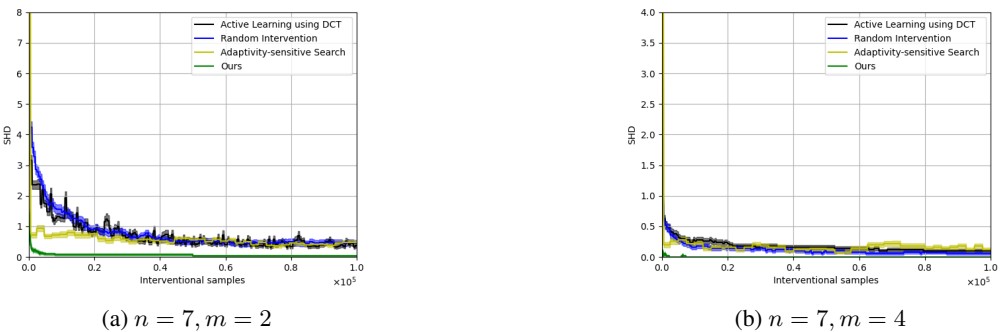

(a) $n = 7, m = 2$             (b) $n = 7, m = 4$

Figure 5: SHD vs number of interventional samples for scale-free graphs generated from Barabási-Albert (BA) model. We generated 50 random DAGs under two settings and plot the average SHD and standard deviation.

## H.3 Additional Experiment on Other Baseline

In this experiment, we compare our proposed algorithm with a Bayesian causal discovery method, AVICI, proposed in Lorch et al. [2022]. For AVICI, we fine-tuned the pretrained models, *scm-v0* and *neurips-rff*, with 50 random complete graphs, each with 1000 observational samples. The results are plotted in Figure 6. We used SHD to measure our algorithm and expected SHD for AVICI. Here, since AVICI can not take in too many samples, we just show the performance with 1000 interventional samples. At each step for both methods, we choose the interventional target randomly. For AVICI, we feed in 1000 observational samples together with the interventional samples. The results show that our algorithm performs better, while AVICI failed to learn the causal structure with high expected SHD. AVICI's bad performance in this setting may be caused by its parametric model on the SCM or noise. Also, there is no theoretical guarantee on AVICI's performance and convergence to the ground truth. Furthermore, our algorithm benefits has the advantage of being able to update the posterior given any number of samples, while AVICI would fail when the number of samples is large.

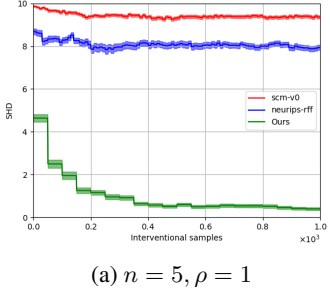

(a) $n = 5, \rho = 1$

Figure 6: SHD vs number of interventional samples for random complete graphs. We generate 50 random DAGs and plot the average SHD and standard deviation. *scm-v0* and *neurips-rff* are pretrained models in the paper.

