# OpenReview forum: "Sample Efficient Bayesian Learning of Causal Graphs from Interventions"
_NeurIPS.cc/2024/Conference — NeurIPS 2024 poster_

### Official Review · Reviewer_HSDU · 2024-07-10

**Soundness:** 3
**Presentation:** 1
**Contribution:** 2
**Rating:** 6
**Confidence:** 2

**Summary:**

This paper introduces a way of doing causal discovery from interventional samples using bayesian inference. They prove some theoretical results of convergence of the task at hand. Finally, they test their approach against common benchmarks on synthetic data.

**Strengths:**

- Apart from the related work section (see weaknesses) the paper is clear and well organized.
- The experiments support the proposed method against the benchmarks.
- The theoretical results are interesting and I believe using the results by Weinöbst et.al. (2023) is a creative solution to decrease the time complexity of the task.

**Weaknesses:**

In my opinion, the related work section is not written very well. It reads like a list with one sentence of what they are doing and not putting your research in the context of previous research. The only exception is line 124.

**Questions:**

I’m still unsure about the overall significance of the paper. Yes, there is section 6 where the authors give a case study for estimating the probability of a set being a valid adjustment set. Furthermore, the authors briefly explain the case of cloud computing and cancer research. However, I don’t understand how that relates to querying interventional data from the system. Could the authors please expand on this? I have not reduced the score from this because I currently believe the strength of the paper are the theoretical results and the approach itself.

**Limitations:**

See questions

---

> ### Author Rebuttal · Authors · 2024-08-07
>
> We thank the reviewers for their thorough and constructive feedback. Below, we address each of the points raised.
>
> **Weakness of Writing:**
>
> We will rearrange the related work section in the revision to position the previous works and our study better.
>
> **Questions of Overall Significance:**
>
> The main contribution of this study is the Bayesian learning algorithm for causal discovery. We use DAG sample to efficiently compute the interventional distributions. With section 6, we want to show that the DAG sampler could be used in more general cases to estimate general causal queries and the case of "a set being a valid adjustment set" is presented as an example. We mentioned real-world scenarios of cloud computing and cancer detection. They can also be transformed into such causal queries without learning the whole graph. To elaborate more, for the root cause analysis in for cloud computing, we could estimate the posterior of the configuration of a vertex to be all outgoing from that vertex to its neighbors. For direct cause analysis in cancer research, we could estimate the posterior of that edge being directed from the potential cause to the cancer variable. Regarding the study of interventional experimental design, there have been plenty of works in the literature (**[1]**, **[2]**, **[3]**, **[4]**, ...), while few of them consider the cases of limited interventional samples. However, these cases are common in real-world settings since interventional samples are usually a lot more costly to retrieve then observational samples. Thus, in this paper we assume the access to observational distribution while limited intervetinoal samples to reflect such scenarios and our proposed algorithm outperforms previous works.
>
> **[1]** *Karthikeyan Shanmugam, Murat Kocaoglu, Alexandros G Dimakis, and Sriram Vishwanath. Learning causal graphs with small interventions. Advances in Neural Information Processing Systems, 28, 2015.*
>
> **[2]** *Alain Hauser and Peter Bühlmann. Two optimal strategies for active learning of causal models from interventional data. International Journal of Approximate Reasoning, 55(4):926–939, 2014.*
>
> **[3]** *Yang-Bo He and Zhi Geng. Active learning of causal networks with intervention experiments and optimal designs. Journal of Machine Learning Research, 9(Nov):2523–2547, 2008.*
>
> **[4]** *Chandler Squires, Sara Magliacane, Kristjan Greenewald, Dmitriy Katz, Murat Kocaoglu, and Karthikeyan Shanmugam. Active structure learning of causal dags via directed clique trees. Advances in Neural Information Processing Systems, 33:21500–21511, 2020.*
>
> We hope that our rebuttal has clarified the reviewer's concerns, and we would be more than happy to engage in further discussions if the reviewer has additional questions.

---

> > ### Comment · Reviewer_HSDU · 2024-08-09
> > **Answer to response**
> >
> > I thank the authors for taking the time to answer to my questions and for taking the suggestion to improve the related work section. Give my current understanding of the paper and the authors' responses I will keep my score as is.

---

### Official Review · Reviewer_i7ee · 2024-07-12

**Soundness:** 3
**Presentation:** 2
**Contribution:** 2
**Rating:** 5
**Confidence:** 3

**Summary:**

This paper proposes a Bayesian approach for learning causal graphs with limited interventional samples. The proposed algorithm first constructs a separating system to design intervention targets and then enumerates the causal effects for all possible cutting edge configurations for each target, and tracks their posteriors. The authors provide theoretical guarantees on the convergence to the true causal graph with sufficient interventional samples, and the experiments on simulated chordal graphs demonstrate that the proposed method requires significantly fewer interventional samples than baselines to achieve low SHD.

**Strengths:**

1. The authors provide a detailed theoretical analysis, proving that the proposed algorithm converges to the true causal graph with high probability given sufficient interventional samples.

2. The related work part is comprehensive and easy to read and understand.

3. Experiments on simulated chordal graphs demonstrate that the proposed method achieves low SHD using significantly fewer interventional samples compared to baselines.

**Weaknesses:**

Major:
1. The proposed method should not be classified strictly as "Bayesian learning of causal graphs," because it does not calculate the posterior distribution over causal graphs directly. Instead, it computes posterior probabilities of specific cutting edge configurations within the graph. Based on these probabilities, the proposed method updates the output DAG accordingly. It would be better if the authors clarified that in the paper to avoid any confusion.

2. It is unclear to me how the proposed method selects intervention targets at each step. I wonder if the authors could further clarify whether the proposed method also designs the intervention targets or the interventions are performed randomly.

3. The authors did not discuss the computational complexity in the paper. The sizes of nodes in the experiments are also relatively small. I wonder if the authors could discuss the complexity of the proposed method and whether it could scale up to graphs with a larger number of nodes.

4. In the experiments, the authors only consider simulated chordal graphs, which naturally fit the proposed method. I think the authors should conduct more experiments on different types of graphs (e.g., scale-free graphs) and perhaps some semi-synthetic graphs (e.g., graphs simulated using realistic simulators [1]).

5. The authors only consider 3 baselines in the experiments, and I wonder if the authors could compare the proposed method with Bayesian causal discovery methods that can handle interventional data (e.g., [2], [3]).


Minor:
1. Missing related work: [3].

2. Please use \citet and \citep correspondingly rather than only using \citet. For example, in line 72: Meek Rules Meek [1995] --> Meek Rules [Meek, 1995]


[1] Dibaeinia, P., & Sinha, S. (2020). SERGIO: a single-cell expression simulator guided by gene regulatory networks. Cell systems, 11(3), 252-271.

[2] Lorch, L., Sussex, S., Rothfuss, J., Krause, A., & Schölkopf, B. (2022). Amortized inference for causal structure learning. Advances in Neural Information Processing Systems, 35, 13104-13118.

[3] Hägele, A., Rothfuss, J., Lorch, L., Somnath, V. R., Schölkopf, B., & Krause, A. (2023, April). Bacadi: Bayesian causal discovery with unknown interventions. In International Conference on Artificial Intelligence and Statistics (pp. 1411-1436). PMLR.

**Questions:**

1. Please see the questions in the Weaknesses part.

2. How is the value of $k$ of the $(n, k)$-separating system determined in the experiments?

3. How is the prior of each configuration defined?

**Limitations:**

The authors adequately addressed the limitations of their work.

---

> ### Author Rebuttal · Authors · 2024-08-07
>
> We thank the reviewers for their thorough and constructive feedback. Below, we address each of the points raised.
>
> **Weakness of Major 1:**
>
> Our approach differs from most of the Bayesian causal discovery paper in that we do not have a parametric representation of the structure. We do not directly calculate the posterior of each DAG since the size of MEC could be large and intractable **[1]**. Instead, we partition the DAGs into different configurations of a given target set to avoid keep tracking of posteriors of all the DAGs in an Bayesian approach. We could explicitly mention this difference in the paper.
>
> **Weakness of Major 2:**
>
> In our implementation, the interventional target is selected randomly. If the maximum posterior of a configuration is close to 1 ($>0.99$ for example), we remove this target to improve efficiency. This target selection process could be made adaptive to further improve the algorithm.
>
> **Weakness of major 3:**
>
> In our work, we assume the access to observational distribution. In our simulation, we use binary variables, so it takes $2^n$ in memory. When the graph is large, it is intractable. For a target of $k$ vertices, in the initialization step where we need to enumerate all the configurations, the complexity would be up to $2^{kd}$ where $d$ is the maximum degree. This can be large when the graph is large and dense. If the graph is large, our algorithm can only work when it is sparse, i.e., the maximum degree is less than $\log n$. However, usually causal graphs are sparse, and can be divided into smaller chain components with observational distribution, which can be oriented independently.
>
> **Weakness of major 4:**
>
> We can also compare the algorithms on other graphs like BA graph. We show the results in the extra pdf in Figure 1. We can see that our algorithm still outperforms others for scale-free graphs. In fact, with the access to the observational distribution, our algorithm does not rely on the type of graphs since the chain components are proved to be chordal **[2]** and can be oriented independently **[3]**. We could not run our algorithm with the mentioned real-world simulator since it does not provide the observational distribution.
>
> **Weakness of major 5:**
>
> We compared with the Avici model in **[4]** and the result if shown in the 1-page pdf of Figure 2. We compare with both 'scm-v0' and 'neurips-rff' model and the result shows that our method performs better. In fact, Avici does not seem to be converging with our simulated data. There could be several reasons for that. First, Avici has no guarantee of performance as mentioned in the paper, thus it might not perform well for some data. Besides, Avici may not work well with discrete data. Also, Avici could not take in too large data since it only loads data once, while our approach has this anytime property to return the optimal DAG with any amount of samples available.
>
> **[5]** is also an important work. As it assume unknown interventions, we did not compare with it here, but we will add it in the related work section in revision.
>
> **Weakness of minor 1, 2:**
>
> We will fix these typos in the revision.
>
> **Question 2:**
>
> In the experiments, we use $k=3$ for small graphs ($n\leq10$) and $k=1$ for $n=20$. When the graph is small, using a slightly larger $k$ could make the algorithm more efficient, but when the graph gets large, we just use atomic interventions to avoid using too much memory.
>
> **Question 3:**
>
> In this study, we assume that at the beginning, all the DAGs in the MEC are equally likely to be the causal graph. Thus, we use equation 7 to calculate the prior of each configuration. More specifically, we divide the MEC size of the configuration by the MEC size of the whole graph. The MEC size could be efficiently calculated using algorithm in **[6]**.
>
> **[1]** *He, Yangbo, Jinzhu Jia, and Bin Yu. "Counting and exploring sizes of Markov equivalence classes of directed acyclic graphs." The Journal of Machine Learning Research 16, no. 1 (2015): 2589-2609.*
>
> **[2]** *Steen A Andersson, David Madigan, and Michael D Perlman. A characterization of markov equiva-389
> lence classes for acyclic digraphs. The Annals of Statistics, 25(2):505–541, 1997.*
>
> **[3]** *Alain Hauser and Peter Bühlmann. Two optimal strategies for active learning of causal models from interventional data. International Journal of Approximate Reasoning, 55(4):926–939, 2014.*
>
> **[4]** *Lorch, L., Sussex, S., Rothfuss, J., Krause, A., and Schölkopf, B. (2022). Amortized inference for causal structure learning. Advances in Neural Information Processing Systems, 35, 13104-13118.*
>
> **[5]** *Hägele, A., Rothfuss, J., Lorch, L., Somnath, V. R., Schölkopf, B., and Krause, A. (2023, April). Bacadi: Bayesian causal discovery with unknown interventions. In International Conference on Artificial Intelligence and Statistics (pp. 1411-1436). PMLR.*
>
> **[6]** *Marcel Wienöbst, Max Bannach, and Maciej Li ́skiewicz. Polynomial-time algorithms for counting
> and sampling markov equivalent dags with applications. Journal of Machine Learning Research, 24(213):1–45, 2023.*
>
> We hope that our rebuttal has clarified the reviewer's concerns and would request that they reconsider their score. We would be more than happy to engage in further discussions if the reviewer has additional questions.

---

> > ### Comment · Reviewer_i7ee · 2024-08-10
> > **Official Comment by Reviewer i7ee**
> >
> > Thank you for the detailed response and the additional experiments. However, I still have some questions regarding the proposed method and the experimental setup.
> >
> > 1. From my understanding, the proposed method consists of two steps: intervention design and causal discovery. It is unclear whether the method is intended to generate higher-quality interventions or to infer the causal graph more accurately with limited interventional samples compared to other causal discovery methods. If the focus is on the former, the method may be more aligned with causal experimental design rather than causal discovery. Could you please elaborate on this point?
> >
> > 2. Regarding the additional experiments, since AVICI is a Bayesian causal discovery method, did you use the expected SHD as the measure? The figures only show SHD, so it would be helpful if you could provide more details about the experimental setup.
> >
> > 3. When comparing with AVICI, did you use the same interventions for both methods? Specifically, how were the interventions for AVICI generated—were they random or derived from the proposed method? Additionally, how did you generate random complete graphs? If pre-trained AVICI models were used, the prediction accuracy could drop significantly if the distribution of the generated graphs differs from the pre-trained set. From Figure 2 in the newly added document, it seems that the SHD of the proposed method is much smaller than that of AVICI at the initial stage. Could you clarify this?
> >
> > 4. Finally, while complexity is a crucial factor in real-world scenarios, the proposed method seems computationally expensive. Additionally, I believe we can obtain observational distributions from real-world simulators, which is similar to synthetic generation (e.g., scale-free). Could you address the computational aspects of your method in this context?
> >
> > I am willing to adjust my score if these concerns are addressed.

---

> > > ### Author Response · Authors · 2024-08-11
> > >
> > > We thank the reviewers for the constructive reply. Below we address each of the points mentioned.
> > >
> > > **Question of objective**
> > >
> > > In our setup, we begin by constructing a set of targets to intervene on. We use an $(n, k)$-separating system here because it is guaranteed to learn every graph in the MEC. During the discovery process, we randomly choose a target from the set, so it is different from the experimental design works where interventional targets are adaptively selected based on previous interventions. The basic purpose of this work is to come up with an anytime algorithm that predicts an optimal graph given limited interventional samples. We will clarify this in the revised manuscript.
> > >
> > > **Question metrics in the additional experiment**
> > >
> > > For the results in Figure 2 of extra experiments, we just use SHD of the prediction as done in Avici's tutorial. More specifically, a threshold of $0.5$ is used to filter the predicted adjacency matrix. We agree that expected SHD would be a better metric to use here for Avici. We can modify this part in revision.
> > >
> > > **Questions about experiment details**
> > >
> > > **Intervention targets:** For both methods, we choose target randomly for each sample. For Avici, we feed 1000 observational samples to the pre-trained models before feeding interventional samples.
> > >
> > > **Graph generation:** We generate the graph following the steps described in section 7, second paragraph. More specifically, here, we first generate a random ordering $\tau$ of $[5]$. Then, we orient $a \rightarrow b$ if $a$ is previous to $b$ in $\tau$ for $a \neq b, a, b \in [5]$. Since the complete graph is chordal, we do not need to perform the last chordalizing step. A DAG $\mathcal{D}$ randomly generated in this way is guaranteed to have a chordal skeleton $G$ and $\mathcal{D} \in [G]$. We do notice that the performance of Avici is highly related to the pre-trained model it is using. We can try to improve Avici's performance by modifying the training part.
> > >
> > > **Initial SHD:** For a complete graph with $5$ vertices, there will be $10$ edges, thus SHD is bounded by $10$. Our method has an initial SHD of $5$, which is basically a random guess of all the edges. However, Avici's metric uses a threshold of $0.5$, which probably masked out most entries in the adjacency matrix, resulting in a high SHD.
> > >
> > > **Question about complexity:**
> > >
> > > We agree with the reviewer that one can obtain observational distributions from real-world simulators, which is similar to synthetic generation. We briefly discuss the computational complexity of our proposed algorithm. Our approach has two major steps. In the first step, we initialize the targets and compute the interventional distributions. In the second step, we merely sample from the intervened Bayes net, put the sample into the interventional distributions, and update the prior and posteriors. The most computationally expensive part is saving the observational and interventional distributions. Since we are using binary variables in our simulation, the joint distribution of $50$ vertices would be a table of $2^{50}$ float numbers, which is intractable. If there is a compact way to represent the joint distributions, our algorithm will still work. We will add this clarification in the revised manuscript.
> > >
> > > We hope we have addressed the questions and would be please to discuss further if reviewer has further concerns.

---

> > > > ### Comment · Reviewer_i7ee · 2024-08-12
> > > > **Official Comment by Reviewer i7ee**
> > > >
> > > > Thank you for the explanation and response. I now understand the main objective of the proposed method.
> > > >
> > > > 1. Regarding the experimental details, I would like to clarify whether "For AVICI, we feed 1000 observational samples to the pre-trained models" means that you fine-tuned the pre-trained AVICI using these 1000 observational samples. If the pre-trained AVICI was directly applied to the simulated graphs without fine-tuning, the poor performance could be explained by a significant difference between the distributions of the simulated graphs and the pre-trained graphs. However, if the pre-trained AVICI was fine-tuned with the observational samples, it is surprising that AVICI's performance is still so poor (a SHD of 8 or 9, which suggests the predictions were almost entirely incorrect).
> > > >
> > > > 2. I am still not very clear about when and how the proposed method can be applied in real-world scenarios. From my understanding, the proposed method is best suited for inferring causal graphs when there are limited intervention samples with known intervention targets. However, as the proposed method requires the observational distributions to be known, I am not sure whether this assumption can be easily fulfilled in real-world applications (e.g., gene regulatory networks). I feel like the case study in Section 6 might be a good application of the proposed method, but the authors did not perform experiments for validation.
> > > >
> > > > 3. I still feel that the experiments should include real-world (or realistic simulated) experiments to demonstrate the effectiveness of the proposed method.
> > > >
> > > > I appreciate the authors' effort and time during the rebuttal, therefore I raised my score by 1. However, I still feel that this paper needs more preparation, especially in describing the main objective, applying the theoretically sound method to real-world applications, and relaxing the assumptions to allow the causal discovery for a larger number of variables.

---

> ### Author Response · Authors · 2024-08-12
>
> We thank the reviewer for increasing our score but would like to briefly address the concerns highlighted by reviewer.
>
> **AVICI experiment details**
>
> In our experiment, we did not fine-tune the pre-trained models using observational samples. We directly feed them with the interventional samples. We will try to fine-tune the pre-trained model first with the observational samples.
>
> **Question of real-world application**
>
> In real-world applications, if we have access to large number of observational samples, we can apply our algorithm on a set of candidate CPDAGs, We will also add a few simple experiments to validate the case study in section 6.
>
> **Experiments with real-world simulators**
>
> We will add experiments using real-world simulators.

---

### Official Review · Reviewer_UxJA · 2024-07-12

**Soundness:** 3
**Presentation:** 3
**Contribution:** 3
**Rating:** 6
**Confidence:** 3

**Summary:**

The paper considers the problem of learning causal graphs using limited interventional samples through the a Bayesian perspective. An algorithm is proposed which returns the most probable causal graph given a limited set of samples. The approach is empirically evaluated and code is given.

**Strengths:**

I did not check the proofs in detail but the approach is sound and experimental results are promising.

**Weaknesses:**

The experiments feels a little "small scale". This is probably due to the need to enumerate all possible configurations via Algorithm 3 for this approach to work (please correct me if I am mistaken).

**Questions:**

- Lines 39-41 discusses the trade-off between non-adaptive and adaptive methods. This was explored in [1], a paper you have already cited. You may want to consider adding a reference to [1] in this section of the introduction in your revision.
- Line 104: Typo of $O(\log n)$?
- Line 104: Is it size 1 or size $n/2$ interventions? For example, see Page 58 of [2] which you cited.
- Line 243 and 514: Do you mean $D_{KL}$? Also, you should define KL divergence properly somewhere in the preliminaries.
- Equation (2): Do you mean $p_j$ in the numerator?
- Line 605: Typo of "Enumerating"?

[1] Davin Choo and Kirankumar Shiragur. "Adaptivity complexity for causal graph discovery." Uncertainty in Artificial Intelligence. PMLR, 2023.

[2] Frederick Eberhardt. "Causation and intervention." Unpublished doctoral dissertation, Carnegie Mellon University 93 (2007).

**Limitations:**

Nil

---

> ### Author Rebuttal · Authors · 2024-08-07
>
> We thank the reviewers for their thorough and constructive feedback. Below, we address each of the points raised.
>
> **Weakness of "small scale" experiments:**
>
> In this work, we assume the access to the joint observational distribution. When the graph is large, it is intractable in practice to handle the joint distribution. For example, in our implementation, we use binary variables. A graph of 50 vertices would lead to a table of $2^{50}$ numbers. Also, to enumerate the configurations of a set of target of $k$ vertices, we have to consider up to $2^{kd}$ configurations in the worst case where $d$ is the maximum degree of the target. This will be huge complexity for large dense graphs, so we just experimented on "small scale" graphs. Although in practice, causal graphs can be large, they are usually sparse and with observational distribution, it can be divided into small chain components and orient independently.
>
> **Question about citations:**
>
> We will add this mentioned paper to this section.
>
> **Questions about typos:**
>
> Thanks for pointing out the typos. We will fix them in revision.
>
> We hope that our rebuttal has clarified the reviewer's concerns, and we would be more than happy to engage in further discussions if the reviewer has additional questions.

---

> > ### Comment · Reviewer_UxJA · 2024-08-10
> >
> > Thank you for your responses. I am still not convinced by the size of the experiments but I do see value in the proposed approach. For instance, I felt it was interesting that they used the method of Wienöbst for efficiency. As such, I will keep my positive score as it is.

---

### Official Review · Reviewer_uDQi · 2024-07-12

**Soundness:** 3
**Presentation:** 2
**Contribution:** 2
**Rating:** 4
**Confidence:** 3

**Summary:**

In this paper, the authors present a Bayesian method to learn the causal graph from observational data and limited interventional data. The authors assume that there are plenty of observational data, which can be used to learn a ground-truth CPDAG. Then, by using the efficient DAG enumeration method to sample DAGs and calculate the posterior, one can find the DAG that is most consistent with the interventional data.

**Strengths:**

In practice, the number of interventional data is limited, thus considering causal discovery with limited data is valuable. The whole idea in this paper is sensible. Despite some missing related studies, the authors give a detailed introduction to some of the relevant studies.

The theoretical results seem interesting, though I do not dig into the details.

**Weaknesses:**

Using Bayesian method to learn causal relations is not novel. There are many existing studies in the literature. Further, from the viewpoint of considering limited samples, it is more sensible to take the uncertain in learning the CPDAG into account.

The writing could be improved. There are a lot of contents in introduction that are better to appear in Related works. Besides, it is not quite clear that what role Section 4.1 play in Section 4: Algorithm Initializations.

It is better to distinguish \citet and \citep.

**Questions:**

It seems that Algorithm 3 could be improved further. If the authors orient the edges of $\mathbf{S}$ one by one and update the graph with Meek rules (due to the completeness of Meek rules in incorporating background knowledge), the complexity will be reduced and there is no need to detect unshielded colliders or cycles. Am I right?

**Limitations:**

No.

---

> ### Author Rebuttal · Authors · 2024-08-07
>
> We thank the reviewers for their thorough and constructive feedback. Below, we address each of the points raised.
>
> **Weakness of Objective:**
>
> There are indeed plenty of studies in the literature that use Bayesian methods for causal discovery, but most of them have parametric assumptions like a linear causal model or additive Gaussian noise (see related works section). These methods fail when the assumptions do not hold. Our approach differs in that we do not make such assumptions, and we show the convergence rate theoretically.
>
> In our study, we assume access to the observational distribution, which naturally leads to the CPDAG. Our work can be extended to the setting of uncertain CPDAGs by updating and tracking the posteriors of the CPDAGs together with the configurations.
>
> **Weakness of writing:**
>
> We will try to improve the writing and clarity of the paper to make it easier to follow. We will move some content from the introduction to the related works section to better position our paper.
>
> Section 4.1 is intended to briefly describe the separating system since it is used in the algorithm and the analysis in Section 5.
>
> **Weakness of citation:**
>
> We will fix the citations.
>
> **Questions:**
>
> In practice, we can list all the configurations and examine them in parallel which is fast since cycle detection and unshielded collider detection algorithms are efficient. The suggested approach saves memory but has to be performed in sequence.
>
> **Additional Thoughts**
>
> While we value constructive criticism that can help us improve our work, we believe that the weaknesses mentioned are generic and can be easily fixed. For instance, comments regarding typos and writing, while valid for copyediting purposes, do not significantly impact the contributions of the paper. The reviewer mentions a lack of novelty as a weakness but does not provide specific points to substantiate this claim. In fact, our work has a clear position, which we state in the introduction and related works section.
>
> A score of 3 suggests significant issues with the paper's contributions, methodology, or results. However, the reviewer does not articulate any major flaws that would warrant such a score. We thus kindly request that the reviewer reconsider the score or provide more justification for the decision to reject our paper. We thus kindly request that the reviewer reconsider the score or provide additional justification for their decision.

---

> > ### Comment · Reviewer_uDQi · 2024-08-13
> > **Thank you**
> >
> > Thank the authors for the response. After reading the rebuttal of the authors and the other reviewers' comments, I think the theoretical result in Thm. 3 is somewhat interesting, thus I increase my score to 4. However, causal discovery using a Bayesian method is not firstly studied. And I agree that using the result of Wienöbst et al. is a nice step, but it is not contributive enough. Hence I insist on my negative score.

---

> ### Author Response · Authors · 2024-08-14
>
> We thank the reviewer for the increasing the score. We want to briefly clarity the concerns below.
>
> Although Bayesian causal discovery is not new, a fully non-parametric Bayesian approach is not tractable. Our work introduces a novel idea by using the $(n, k)$-separating system, followed by an enumeration of causal effects to partition the set of all possible DAGs into tractable sets. At the prediction stage, we combine the configurations with high posterior to return a DAG. Furthermore, we show in theory that our method is guaranteed to converge to the true DAG as sample size $m\rightarrow \infty$ in Lemma 1. Additionally, we show that when the sample size is large, the predicted graph would be the true graph with a high probability as shown in Theorem 3. This is the first work to show such guarantees in the context of sample-efficient causal discovery problems under mild faithfulness assumptions.

---

### Author Rebuttal · Authors · 2024-08-07

**Extra Experiments**

We included the results of extra experiments in the pdf as required by the reviewers.

**Experiment for Scale-Free Graphs**

We generate 50 random scale-free graphs under 2 setting: $n=7, m=2$ and $n=7, m=4$, and compare with the baselines. The results are plotted in Figure 1. Our proposed algorithm outperforms other methods.

**Experiment to Compare with Avici**

Avici is a Bayesian causal discovery model proposed in **[1]**. We compare our method with 2 pretrained models "scm-v0* and *neurips-rff" provided by the authors. Both models are trained on non-linear SCMs. We generated 50 random complete DAGs with 5 vertices. Since Avici cannot take in large data, we just compare the algorithms on 1000 interventional samples. The result is shown in Figure 2. Our algorithm outperforms both Avici models.

**[1]** *Lorch, L., Sussex, S., Rothfuss, J., Krause, A., and Schölkopf, B. (2022). Amortized inference for causal structure learning. Advances in Neural Information Processing Systems, 35, 13104-13118.*

---

### Author Response · Authors · 2024-08-14
**Final Thoughts**

We thank all reviewers for providing thoughtful and constructive feedback. In response, we have carefully considered each of the points raised and made significant revisions to the manuscript to address the concerns. Specifically, we added extra experiments required by Reviewer i7ee to test the performance of proposed algorithm on scale-free graphs and compare our proposed algorithm with AVICI proposed in **[1]**. In the revision, we will do the following changes suggested by the reviewers:

- We will also fix the typos and citation formats raised by the reviewers.
- We will take the suggestions on writing by Reviewer uDQi and HSDU.
- We will clarify the objective of this work as suggested by Reviewer i7ee.
- We will improve the performance of AVICI, add experiments to validate section 6, and include experiments using real-world simulators as suggested by Reviewer i7ee.

Here, we briefly conclude our work. In this work, we propose a novel sample-efficient causal discovery algorithm using a Bayesian approach. It is well-known that a fully non-parametric Bayesian approach would be intractable as the MEC size can be large. Previous Bayesian approaches in the literature have some parametric assumptions such linear SCM or additive Gaussian noise. Such methods will fail when the assumption do not hold. To handle this, our proposed algorithm uses $(n, k)$-separating system as interventional targets and further partition the DAGs according to the edge configuration of the target set, which makes the Bayesian learning tractable. Our algorithm also has the anytime property, i.e., predict an optimal graph with any amount of interventional samples. Additionally, we show in theory that our proposed algorithm is guaranteed to converge to the true graph both asymptotically and non-asymptotically.  In section 6, we briefly show that our method could be extended to estimate more general causal queries efficiently using DAG sampler proposed in **[2]**.

We believe these changes will strengthen the manuscript and clarify the contributions of our work. We appreciate the opportunity to improve our paper and are confident that the revised version better reflects the significance and rigor of our research.

Thank you again for your time and effort in reviewing our submission. We look forward to your final decision.

 **[1]** *Lorch, Lars, Scott Sussex, Jonas Rothfuss, Andreas Krause, and Bernhard Schölkopf. "Amortized inference for causal structure learning." Advances in Neural Information Processing Systems 35 (2022): 13104-13118.*

**[2]** *Wienöbst, Marcel, Max Bannach, and Maciej Liśkiewicz. "Polynomial-time algorithms for counting and sampling Markov equivalent DAGs with applications." Journal of Machine Learning Research 24, no. 213 (2023): 1-45.*

---

### Decision · Program_Chairs · 2024-09-25

**Decision:**

Accept (poster)

**Comment:**

This work presents Bayesian approaches to the challenging problem of active learning of causal graphs via interventions. Reviewers were overall positive and I recommend acceptance. That said, reviewers did raise issues with various aspects of the writing - authors please do incorporate the recommendations and address the comments as promised in the rebuttal.